# IL-1β promotes adipogenesis by directly targeting adipocyte precursors

Kaisa Hofwimmer[1,6], Joyce de Paula Souza[2,3,6], Narmadha Subramanian[1], Milica Vujičić [4], Leila Rachid[2,3], Hélène Méreau[2,3], Cheng Zhao[2,3], Erez Dror[2,3], Emelie Barreby [5], Niklas K. Björkström [5], Ingrid Wernstedt Asterholm [4], Marianne Böni-Schnetzler [2,3], Daniel T. Meier [2,3,6] ✉, Marc Y. Donath [2,3,6] & Jurga Laurencikiene [1,6] ✉

Postprandial IL-1β surges are predominant in the white adipose tissue (WAT), but its consequences are unknown. Here, we investigate the role of IL-1β in WAT energy storage and show that adipocyte-specific deletion of IL-1 receptor 1 (IL1R1) has no metabolic consequences, whereas ubiquitous lack of IL1R1 reduces body weight, WAT mass, and adipocyte formation in mice. Among all major WAT-resident cell types, progenitors express the highest *IL1R1* levels. In vitro, IL-1β potently promotes adipogenesis in murine and human adipose-derived stem cells. This effect is exclusive to early-differentiation-stage cells, in which the adipogenic transcription factors C/EBPδ and C/EBPβ are rapidly upregulated by IL-1β and enriched near important adipogenic genes. The pro-adipogenic, but not pro-inflammatory effect of IL-1β is potentiated by acute treatment and blocked by chronic exposure. Thus, we propose that transient postprandial IL-1β surges regulate WAT remodeling by promoting adipogenesis, whereas chronically elevated IL-1β levels in obesity blunts this physiological function.

White adipose tissue (WAT) is the central organ for handling and safely storing excess energy, thereby protecting the rest of the body from ectopic lipid accumulation and lipotoxicity. It achieves this task by expanding in two ways, either by enlargement of already existing adipocytes (hypertrophy), or by the formation of new adipocytes (hyperplasia). A hypertrophic WAT (few and large fat cells) has been linked to chronic inflammation, tissue dysfunction, and metabolic diseases, whereas a hyperplastic WAT (many and small fat cells) is associated with protection from metabolic complications[1–9].

Since mature adipocytes are post-mitotic cells, hyperplastic expansion occurs through differentiation of WAT-resident precursors[10]. This process, known as adipogenesis, is tightly controlled by several transcriptional regulators[11]. Briefly, upon in vitro adipogenic induction, a first wave of transcription factors is activated, including CCAT/enhancer-binding protein β (C/EBPβ), C/EBPδ, glucocorticoid receptor, and cAMP response element-binding protein (CREB)[12–16]. These early adipogenic regulators are driving the transactivation and recruitment of a second wave of transcription factors, involving C/EBPα and peroxisome proliferator-activated receptor γ (PPARγ), which in turn initiate the transcriptional program required for final differentiation and mature adipocyte function[15–18].

While major intracellular mediators responsible for activating and relaying the adipogenic program in vitro have been identified, the physiological external signals inducing this process in vivo remain

[1]Lipid Laboratory, Unit of Endocrinology, Department of Medicine Huddinge, Karolinska Institutet, SE-141 52 Huddinge, Sweden. [2]Department of Biomedicine, University of Basel and University Hospital Basel, 4031 Basel, Switzerland. [3]Clinic of Endocrinology, Diabetes and Metabolism, University Hospital Basel, 4031 Basel, Switzerland. [4]Department of Physiology/Metabolic Physiology, Institute of Neuroscience and Physiology, The Sahlgrenska Academy at University of Gothenburg, SE-405 30 Gothenburg, Sweden. [5]Center for Infectious Medicine, Department of Medicine Huddinge, Karolinska Institutet, Karolinska University Hospital, SE-141 52 Huddinge, Sweden. [6]These authors contributed equally: Kaisa Hofwimmer, Joyce de Paula Souza, Daniel T. Meier, Marc Y. Donath, Jurga Laurencikiene. ✉e-mail: daniel.zeman@unibas.ch; jurga.laurencikiene@ki.se

largely unknown. Interestingly, although the chronic WAT inflammation associated with obesity has adverse effects on metabolic health, some local inflammatory signals are essential for maintained metabolic homeostasis. Blocking major inflammatory pathways specifically in adipocytes impairs healthy, hyperplastic WAT remodeling[19,20], and leads to adipocyte death[20], fatty liver[21], and local and systemic insulin resistance[20,21]. Furthermore, an inflammatory surge has been observed in both human and murine WAT postprandially, both in obesity and non-obesity[22–26], suggesting postprandial inflammation is a physiological rather than pathological response to caloric influx, and distinct from the obesity-associated chronic low-grade inflammation.

A major pro-inflammatory factor involved in metabolic regulation is interleukin 1 β (IL-1β). Recently, this cytokine was implicated in adipocyte dysfunction and cell death upon SARS-Cov-2 infection[27]. IL-1β is part of the obesity-associated chronic inflammation that can be detrimental to metabolic health[28]. In this context, its role in the development of type 2 diabetes through induction of peripheral insulin resistance and by direct interaction with pancreatic β-cells, leading to suppression of insulin secretion, dysfunction and cell death has been well described[29–35]. However, in a more acute, transient setting, like postprandially, IL-1β has the opposite effect, as it stimulates insulin secretion from β-cells[32,36–40]. In fact, postprandial IL-1β is intimately involved in glucose homeostasis by enhancing both glucose-induced insulin secretion as well as insulin-induced glucose uptake into immune cells and WAT[37]. Interestingly, transcriptomics of different tissue-resident macrophages isolated postprandially revealed that the IL-1 pathway is upregulated uniquely in WAT-resident macrophages in response to refeeding[26]. In line with this, hyperglycemia has been shown to induce IL-1β production in WAT[41]. Noteworthy, many of these findings were made in healthy, lean, chow-fed mice, suggesting that the postprandial IL-1β elevation in WAT could be part of a natural regulation of energy handling.

Thus, we hypothesized that IL-1β has a physiological role in hyperplastic adipose tissue expansion and fat cell metabolism. To investigate this, we used in vitro and in vivo models to activate or block IL-1 signaling in adipocytes and their precursors and found that IL-1β is of minor relevance for mature healthy adipocytes but plays a crucial role in differentiation of progenitor cells.

## Results

### Deletion of the IL-1 receptor 1 in mature adipocytes prevents acute IL-1β-induced glucose uptake but has no chronic metabolic effect

To study the metabolic consequences of IL-1 system activation in mature adipocytes, we generated adipocyte-specific IL-1 receptor 1 knockout (IL1R1^AKO) mice by breeding IL1R1 floxed mice (IL1R1^FF) to mice expressing Cre recombinase under the control of the adiponectin promoter. The knockout efficiency of *Il1r1* in purified gonadal and subcutaneous adipocytes was 92% and 73%, respectively. *Il1r1* mRNA was unchanged in the stromal vascular fraction of gonadal (gWAT) and subcutaneous WAT (scWAT), confirming the cell-type specificity of the mouse model (Fig. 1a, b). Despite the notable ablation of *Il1r1* in mature adipocytes of chow diet-fed IL1R1^AKO mice, no major changes were observed in mRNA levels of classical adipocyte or inflammatory markers, body weight development, food-intake, glucose, insulin, or IL-1β levels during fasting-refeeding experiments, insulin- or glucose tolerance (with or without IL-1β challenge), adipocyte area, or tissue (including fat pads) weight compared to littermate controls (Fig. 1c-h and Supplementary Fig. 1a–x). Next, we metabolically challenged female and male mice by high-fat diet (HFD) feeding starting at 8–10 weeks of age. IL1R1^AKO mice showed body weight, glucose tolerance, insulin levels, adipocyte area and mRNA markers, plasma glycerol, and tissue (including fat pads) weight similar to the littermate controls (Fig. 1e, i-k and Supplementary Fig. 2a–k). At 21–22 weeks of age, insulin tolerance was slightly improved in male IL1R1^AKO mice but this

transient increase in insulin sensitivity vanished 12 weeks later (Supplementary Fig. 2l, m). Next, we investigated the more immediate response to HFD feeding. Similar to previously reported results[37], acute IL-1β injection increased glucose uptake in gWAT and mesenteric WAT (mWAT) of 12-week-old mice HFD-fed for 4 days (Fig. 1l, m). This effect was not observed in gWAT and it was blunted in mWAT of IL-1β-treated IL1R1^AKO mice, compared to IL-1β-treated control mice. Moreover, there was a trend towards increased basal glucose uptake in IL1R1^AKO gWAT, while insulin-stimulated uptake was similar between genotypes. There was no apparent effect of IL-1β treatment in scWAT (Fig. 1n). Given that adipocytes in scWAT and gWAT express comparable levels of *Il1r1* (Supplementary Fig. 2n), the reduced responsiveness to IL-1β observed in scWAT may be due to a lower proportion of adipocytes in scWAT compared to gWAT[42]. The effect of IL-1β on acute glucose uptake was further examined in an additional assay in wild-type (WT) mice. To avoid potential confounding effects of insulin, these mice were euthanized before an IL-1β-stimulated increase in insulin secretion (Supplementary Fig. 2o). IL-1β injection increased glucose uptake in gWAT also in this model (Supplementary Fig. 2p), although the effect in isolated adipocytes did not reach significance (Supplementary Fig. 2q). Furthermore, we observed an increased glucose uptake by acute IL-1β treatment in human in vitro differentiated adipocytes (Supplementary Fig. 2r). Collectively, these data suggest that IL-1 signaling in mature adipocytes induces acute glucose uptake in WAT without long-term consequences for WAT morphology and whole-body glucose homeostasis.

### Ubiquitous IL1R1 deficiency reduces body weight gain and alters WAT morphology

Considering that mice with adipocyte-specific IL1R1 deficiency did not show a strong metabolic phenotype, we speculated whether other WAT-resident cells could be the major targets of IL-1β secreted from adipose tissue macrophages[26]. To investigate how the absence of IL-1 signaling affects WAT expansion, we examined whole-body IL1R1-knockout (KO) mice and WT controls. Despite comparable food intake (Supplementary Fig. 3a), body weight gain of IL1R1-KO mice on chow diet and HFD was reduced (Supplementary Fig. 3b and Fig. 2a, b). The scWAT mass of IL1R1-KO mice was 61% and 45% reduced after chow diet and HFD, respectively (Fig. 2c), whereas the gWAT mass was 49% and 18% smaller after chow diet and HFD, respectively (Fig. 2d). Histological analysis showed decreased adipocyte size in the scWAT of IL1R1-KO mice, irrespective of diet (Fig. 2e, f), and in the gWAT under chow diet (Fig. 2g). Despite the reduced gWAT mass, the adipocyte size in the gWAT of HFD-fed IL1R1-KO mice was comparable to WT control mice (Fig. 2h). In addition, IL1R1-KO mice showed alterations in the proportions of adipose tissue macrophages, dendritic cells, and eosinophils that might be coupled to body weight regulation (Supplementary Fig. 3c, d). Next, we investigated if a transient pharmacological systemic inhibition of IL-1 signaling in WT HFD-fed mice alters WAT morphology. For this, saline or anakinra (IL-1R1 antagonist, IL-1Ra) was injected daily for 2 weeks starting 2 days before HFD-feeding. After 9 weeks, IL-1Ra-treated mice displayed reduced body weight, WAT mass, and adipocyte size in both gWAT and scWAT (Fig. 2i-n). These results show that, unlike adipocyte-specific IL1R1 ablation, genetic and pharmacological blockade of IL1R1 reduces body weight gain, WAT mass, and adipocyte size.

### IL-1 signaling regulates HFD-induced differentiation of adipocyte progenitor cells in vivo

To get insight into the *Il1r1* expression pattern in WAT, we separated adipose tissue cell compartments and found *Il1r1* to be mainly expressed in the CD45-negative fraction and less prominently expressed in immune cells (CD45-positive) and mature adipocytes (Fig. 2o, p). As the majority of the CD45-negative fraction in WAT contains adipocyte progenitors, we further assessed the role of IL-1

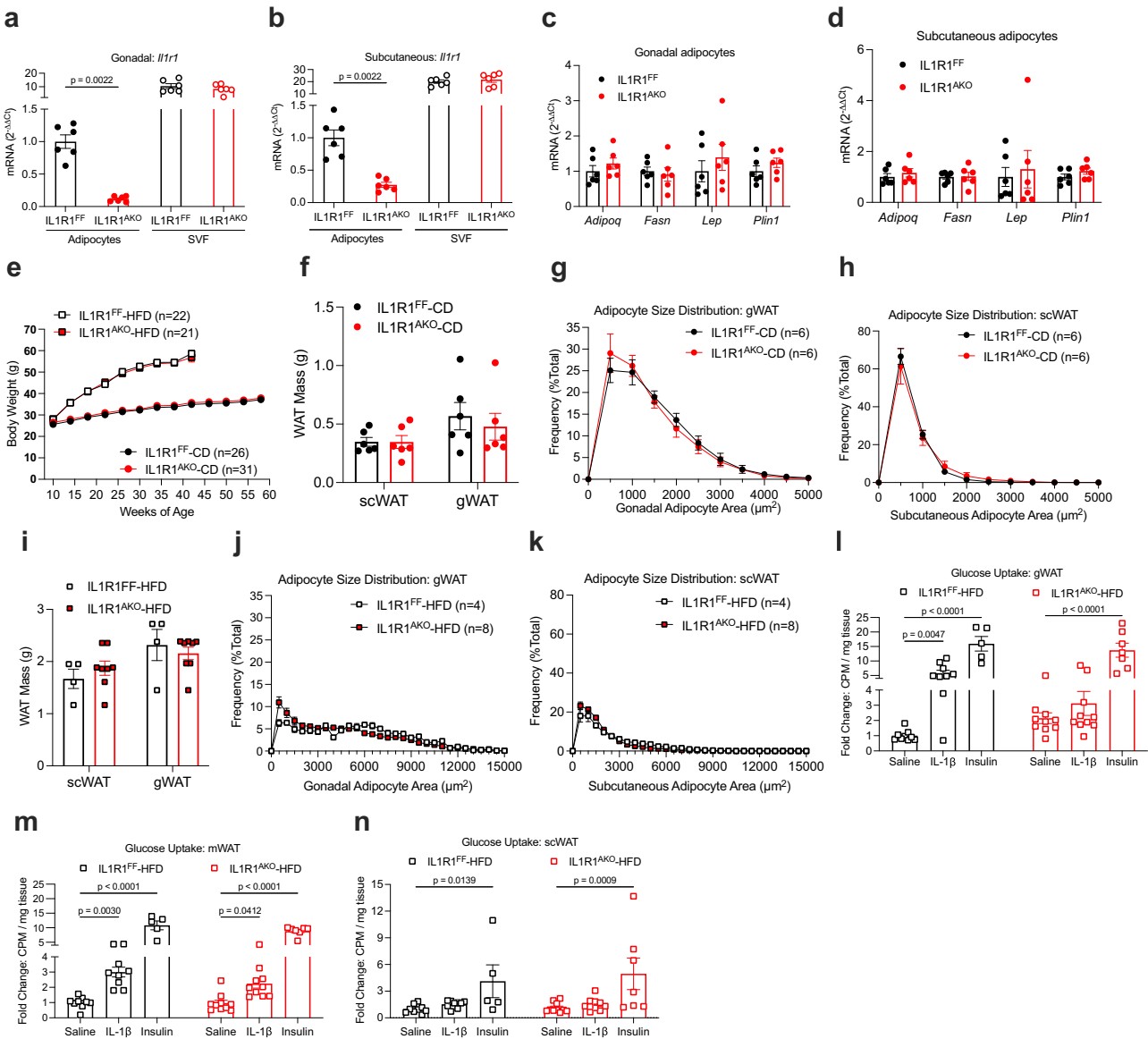

**Fig. 1 | IL1R1^AKO mice have no metabolic phenotype. a, b** Relative mRNA expression in adipocytes and stromal vascular fraction (SVF) isolated from gWAT (**a**) and scWAT (**b**) of 12-week-old chow-diet-fed male mice (n = 6). **c, d** Relative mRNA expression in adipocytes isolated from gWAT (**c**) and scWAT (**d**) of 12-week-old chow-fed male mice (n = 6). **e** Body weight development of chow-fed (IL1R1^FF n = 26; IL1R1^AKO n = 31) and HFD-fed (IL1R1^FF n = 22; IL1R1^AKO n = 21) male mice. **f–h** Fat pad mass (**f**) and adipocyte size distribution in gWAT (**g**) and scWAT (**h**) of chow-fed 17-week-old male mice (n = 6). **i–k** Fat pad mass (**i**) and adipocyte size distribution in gWAT (**j**) and scWAT (**k**) of 17-week-old male mice HFD- fed for 9 weeks (IL1R1^FF n = 4; IL1R1^AKO n = 8). **l–n** In vivo glucose uptake in gWAT (**l**), mWAT (**m**), and scWAT (**n**) of 12-week-old male mice, HFD-fed for 4 days, treated with saline, IL-1β (1 µg/kg bw), or insulin (1U/kg bw). IL1R1^FF: saline (n = 9), IL-1β (n = 9), insulin (n = 5); IL1R1^AKO: saline (n = 9), IL-1β (n = 10), insulin (n = 7). Data are shown as fold change of counts per minute (CPM). n=biological replicates. Data are shown as individual measurements and mean ± SEM. Statistical analyses were performed by unpaired nonparametric two-tailed Mann-Whitney U test (**a–d, f, i**) or two-way ANOVA and Šidák's (**e, g, h, j, k**) or Fisher's LSD (**l–n**) multiple comparison test. Source data are provided as a Source Data File.

signaling in progenitor differentiation. Because new adipocytes arise from progenitors that proliferate prior to differentiation[43–45], we performed in vivo 5-Ethynyl-2′-deoxyuridine (EdU) tracing experiments. 8-week-old mice were treated with EdU for one week, and EdU incorporation in the nuclei of mature adipocytes was quantified 8 weeks later (Fig. 2q). No difference was detected in chow diet-fed EdU-treated mice (Fig. 2r, s). As previously reported, HFD-feeding induced EdU incorporation[44] and this effect was reduced in IL1R1-KO mice (Fig. 2r, s), suggesting that adipocyte progenitor differentiation as an early response to caloric excess is dependent on IL1R1. No major changes in abundance of adipocyte progenitor populations were observed in IL1R1-KO mice (Supplementary Fig. 3e–h). Further, we found that transient pharmacological inhibition of IL-1 signaling,

starting at 8 weeks of age, increased EdU incorporation in gonadal, but not subcutaneous adipocytes of HFD-fed mice (Fig. 2t–v). These results suggest that life-long deletion of IL1R1 impairs, while transient adult-onset IL1R1 blockage promotes adipocyte progenitor differentiation.

## Reduced IL-1β signaling is associated with larger adipocytes in human scWAT

To substantiate the importance of IL-1β signaling in WAT morphology, we next analyzed scWAT gene expression and morphology data from a clinical cohort of healthy women with and without obesity[46]. Expression of both *IL1B* and *IL1RN* (the gene encoding IL-1Ra) were positively associated to body mass index (BMI), body fat percent, and adipocyte

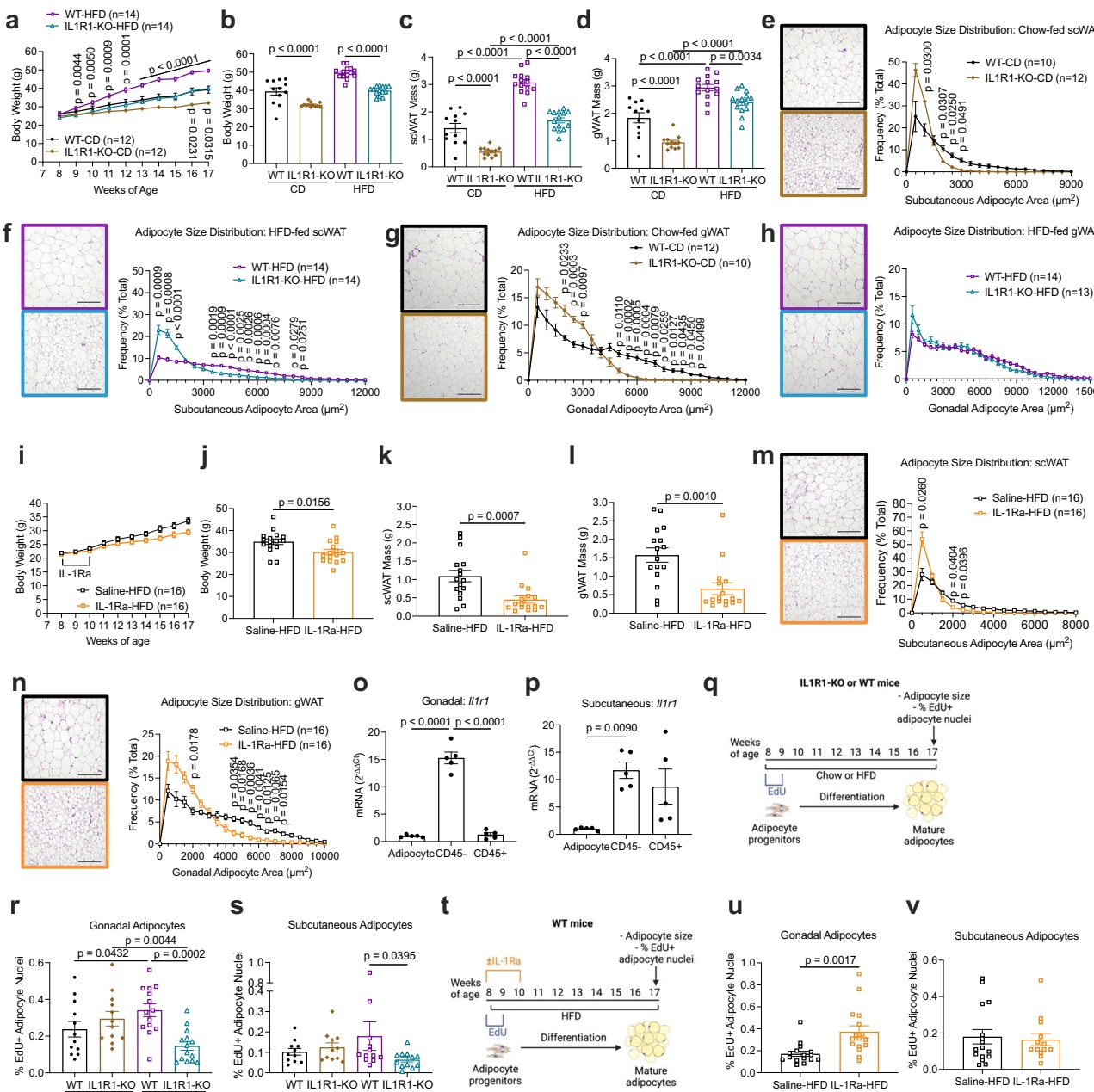

**Fig. 2 | IL-1R1 signaling modulates body weight, WAT mass, adipocyte size, and new fat cell formation. a–d** Body weight development (**a**), body weight (**b**), fat pad mass (**c**, **d**) of 17-week-old chow and HFD-fed male mice. n = 12 WT-CD, n = 12 IL1R1-KO-CD, n = 14 WT-HFD, n = 14 IL1R1-KO-HFD. **e–h** Adipocyte size distribution (**e–h**) of 17-week-old chow and HFD-fed male mice. n = 10 IL1R1-WT-CD, n = 12 IL1R1-KO-CD (**e**). n = 14 per genotype (**f**). n = 12 IL1R1-WT-CD, n = 10 IL1R1-KO-CD (**g**). n = 14 IL1R1-WT-HFD, n = 13 IL1R1-KO-HFD (**h**). **i–n** Body weight development (**i**), body weight (**j**), fat pad mass (**k**, **l**) and adipocyte size distribution (**m**, **n**) of 17-week-old HFD-fed male mice treated with IL-1Ra (10 mg/kg bw daily for 14 days), n = 16 for both conditions. **o**, **p** Relative *Il1r1* mRNA expression in adipocytes, CD45⁻, and CD45⁺ stromal vascular cells isolated from gWAT (**o**) and scWAT (**p**) of 12-week-old chow-fed male WT mice (n = 5). **q–s** Experimental scheme of EdU tracing experiments (100 µg EdU/day, every 2 days, total of 4 injections) (**q**) and percentage of EdU⁺ adipocyte nuclei isolated from gWAT (**r**) and scWAT (**s**) of IL1R1-KO mice and

WT controls. gWAT IL1R1-KO and gWAT WT: chow diet (n = 12 per genotype); HFD (n = 14 per genotype). scWAT IL1R1-KO and scWAT WT: chow diet (n = 11 per genotype); HFD (n = 12 per genotype). **t–v** Experimental scheme of EdU tracing experiments (100 µg EdU/day, every 2 days, total 4 injections) and IL-1Ra therapy (10 mg/kg bw daily for 14 days) (**t**) and percentage of EdU⁺ adipocyte nuclei isolated from gWAT (**u**) and scWAT (**v**) of HFD-fed WT mice. gWAT: saline and IL-1Ra (n = 16). scWAT: saline (n = 16); IL-1Ra (n = 13). Scale bar = 200 µm. Experimental schemes created with biorender.com. n = biological replicates. Data are shown as individual measurements and mean ± SEM. Statistical analyses were performed by: unpaired nonparametric two-tailed Mann-Whitney U test (**j–l**, **u**, **v**); one-way ANOVA and Šidák's multiple comparison test (**o**, **p**); or two-way ANOVA and Šidák's (**a**, **e–i**, **m**, **n**) or Fisher's LSD (**b–d**, **r**, **s**) multiple comparison test. Source data are provided as a Source Data File.

volume (Fig. 3a and Supplementary Fig. 4a–e). After adjusting for BMI or body fat percent, only *IL1RN* expression remained positively significantly associated to fat cell volume (Table 1). Additionally, we examined the relationship between the IL-1β system and the

morphology value[47], which reflects how much the adipocyte size of an individual deviates from what is expected based on their body fat percent, with negative and positive values indicating a hyperplastic and hypertrophic WAT, respectively. The morphology value did not

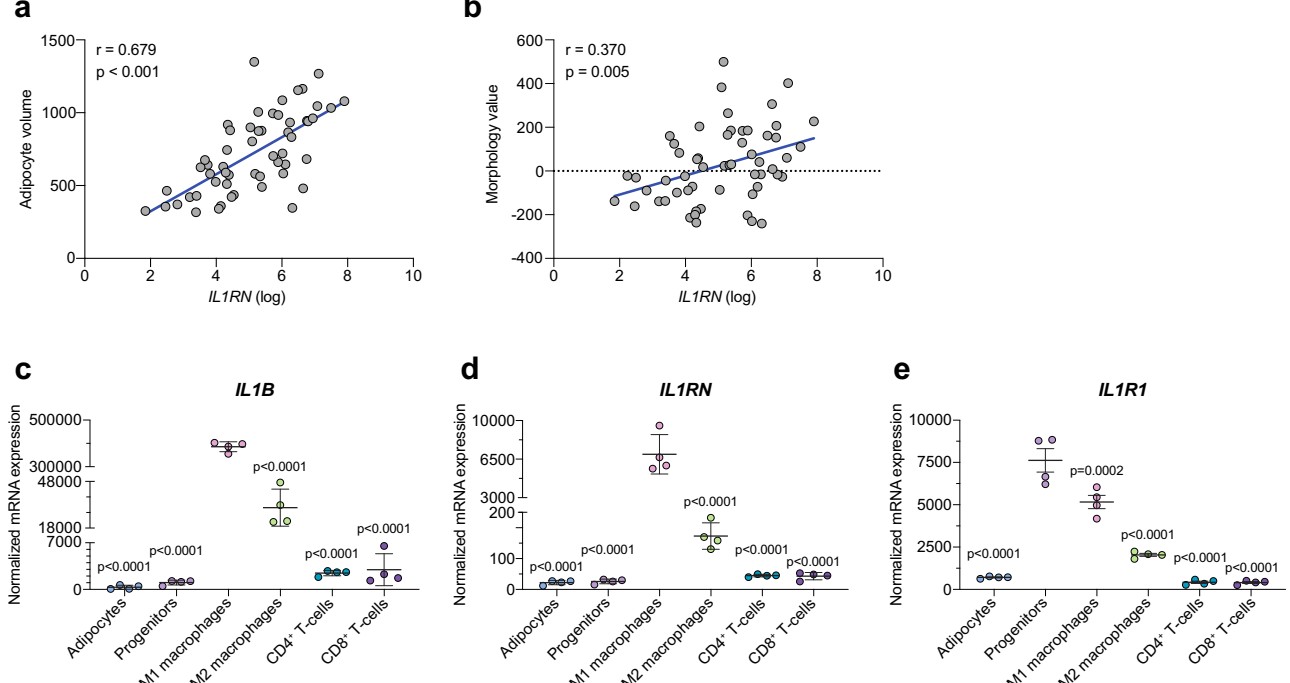

**Fig. 3 | Reduced IL-1 signaling correlates to hypertrophic adipocytes, and progenitors express the highest levels of *IL1R1* in human scWAT. a, b** Pearson correlation of human scWAT *IL1RN* expression and adipocyte volume (**a**) or morphology value (**b**) where positive and negative values indicate WAT hypertrophy and hyperplasia, respectively (n = 56). Two-tailed *p*-values. **c-e** Expression of *IL1B*

(**c**), *IL1RN* (**d**) and *IL1R1* (**e**) in adipocytes and FACs-sorted stromal vascular cell populations from human scWAT (n = 6 donors, pooled 3 and 3 and processed in duplicates). Statistical analyses by one-way ANOVA and Dunnett's multiple comparisons test compared to M1 macrophages (**c**, **d**) or progenitors (**e**). Line and error bars represent mean ± SEM. Source data are provided as a Source Data File.

correlate to *IL1B* expression (Supplementary Fig. 4f) but was positively associated to *IL1RN* expression (Fig. 3b). Taken together, these results suggest that a reduced IL-1β signal is associated with hypertrophic adipocytes in human scWAT.

**Human adipocyte progenitors express the highest levels of IL1R1**
We next analyzed mRNA expression in cell populations of scWAT from human biopsies[48]. Interestingly, while *IL1B* and *IL1RN* were mainly expressed by M1-like macrophages (Fig. 3c, d), the highest expression of *IL1R1* was observed in adipocyte progenitors, which had more than 10-fold higher mRNA levels than mature adipocytes (Fig. 3e). This suggests that, in accordance with murine data presented above, progenitor cells are the main responders to IL-1β in human scWAT.

**IL-1β, but not other inflammatory factors, potently promotes adipogenesis of adipose-derived stem cells**
To test whether IL-1β is involved in differentiation of adipocyte precursors, as suggested by our mouse and human data, we sorted human primary scWAT progenitors by fluorescence-activated cell sorting (FACS) and treated them with IL-1β during their in vitro differentiation.

**Table 1 | Relationship between adipocyte volume and the IL-1β system, adjusted for body mass index or body fat percent, in scWAT from 56 women with and without obesity**

|  | Multiple regression[a] | | Multiple regression[b] | |
|---|---|---|---|---|
|  | **β** | ***p*** | **β** | ***p*** |
| *IL1B* | 0.039 | 0.695 | 0.046 | 0.641 |
| *IL1RN* | 0.342 | 0.001 | 0.344 | <0.001 |

Multiple regression analyses with adipocyte volume as dependent variable and [a]body mass index or [b]body fat percent together with mRNA expression of *IL1B* or *IL1RN* as independent variables.

Lipid droplet quantification in differentiated adipocytes showed that IL-1β strongly stimulated adipogenesis and/or lipid accumulation (Fig. 4a, b), whereas cell number remained unaltered (Supplementary Fig. 5a). Further, in human adipose-derived stem cells (hASCs), IL-1β promoted both lipid accumulation and adipogenic gene expression in a concentration- (Fig. 4c–e) and IL-1R1-dependent (Fig. 4f, g) manner, confirming the adipogenic effect of IL-1β in a second model of primary human cells. In this model, IL-1β treatment also slightly increased cell number (Supplementary Fig. 5b). A more detailed image analysis at single-cell resolution showed that IL-1β markedly shifted the cell distribution towards an increased lipid content, with a striking decrease in the proportion of undifferentiated cells (i.e. the proportion of cells with no or very few lipid droplets) (Fig. 4h, i), further supporting that IL-1β stimulates adipogenesis. Contrastingly, treatment with interferon γ (IFN-γ) and tumor necrosis factor α (TNF-α) decreased lipid droplet formation, while lipopolysaccharide (LPS) and monocyte chemoattractant protein-1 (MCP-1) had no effect (Fig. 4j). IL-6 treatment slightly increased lipid accumulation, but to a significantly smaller extent than IL-1β (Fig. 4j). Next, we investigated whether IL-1β also stimulates in vitro adipogenesis of murine cells. In CD45-negative stromal vascular cells from lean mice, lipid droplet accumulation in response to IL-1β was increased in differentiating cells from scWAT but reduced in gWAT-derived cells (Supplementary Fig. 5c), whereas cell number was slightly increased in cells from both depots (Supplementary Fig. 5d). Similarly, in gWAT-derived cells from ob/ob mice, IL-1β reduced lipid droplet formation and increased cell number (Supplementary Fig. 5e, f). However, in contrast to cells from lean mice, IL-1β had no effect on lipid droplet accumulation or cell number in scWAT-derived cells (Supplementary Fig. 5e, f). Taken together, these results suggest that IL-1β, but not other inflammatory factors, potently promotes adipogenesis of human adipose-derived stem cells and that in murine precursors this effect is restricted to the subcutaneous depot of lean mice.

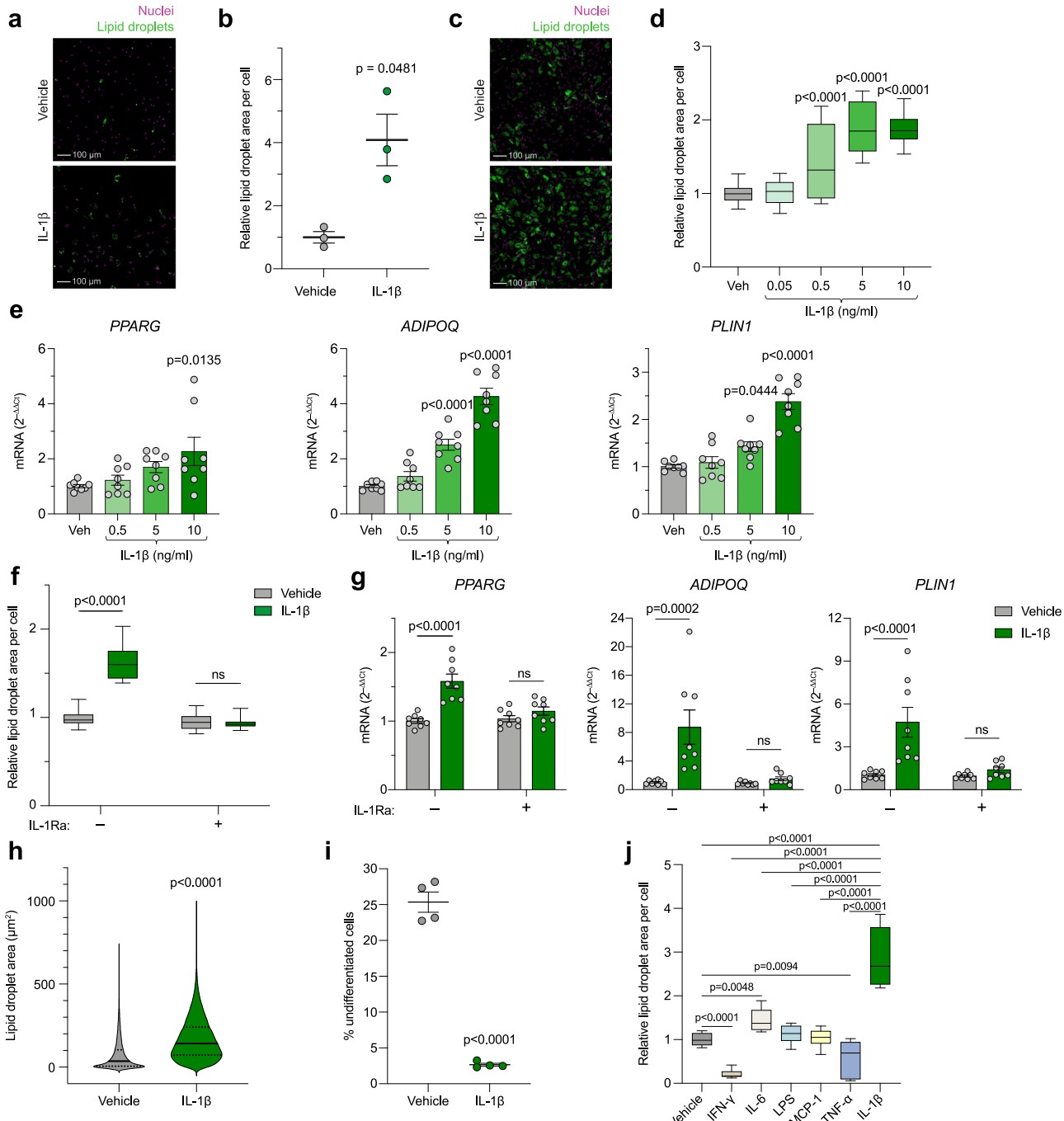

**Fig. 4 | IL-1β, but not other inflammatory factors, potently promotes adipogenesis in human adipocyte progenitors in vitro. a, b** Representative images (**a**) and quantification (**b**) of lipid droplet accumulation in freshly FACS-sorted scWAT-derived human primary adipocyte progenitors differentiated for 13 days with IL-1β (5 ng/ml) (n = 3 donors). Scale bar = 100 μm. **c, d** Representative images (**c**) and quantification (**d**) of lipid droplet accumulation in hASCs differentiated for 9 days with indicated concentrations of IL-1β (10 ng/ml in **c**) (n = 3 independent experiments with ≥ 4 replicates (34 datapoints for vehicle and 12 for the rest)). Scale bar = 100 μm. **e** Adipogenic gene expression on day 2 (*PPARG*) or 5 (*ADIPOQ* and *PLIN1*) of differentiation in hASCs treated with IL-1β from start of differentiation (n = 4 independent experiments with 2 replicates). **f** Lipid droplet accumulation in hASCs differentiated for 9 days with IL-1β (10 ng/ml) and IL-1Ra (500 ng/ml) (n = 4 independent experiments with ≥ 4 replicates (33 datapoints for −IL-1Ra +vehicle and 16 for the rest)). **g** Adipogenic gene expression in hASCs 2 days after adipogenic induction +/− IL-1β (10 ng/ml) and IL-1Ra (500 ng/ml) (n = 4 independent

experiments with 2 replicates). **h, i** Single-cell resolution image analysis of lipid droplet area in individual cells (hASCs differentiated for 9 days with 10 ng/ml IL-1β). Distribution of total area covered by lipid droplets within individual cells (**h**) and % of undifferentiated cells (**i**) (n = 4 analyzed wells per condition, each well containing around $10^4$ cells). **j** Lipid droplet accumulation in hASCs differentiated for 9 days with IFN-γ (10 ng/ml), IL-6 (10 ng/ml), LPS (100 ng/ml), MCP-1 (20 ng/ml), TNF-α (10 ng/ml), IL-1β (10 ng/ml), or vehicle (n = 3 independent experiments with ≥ 4 replicates (14 datapoints for vehicle and 12 for the rest)). Statistical analyses by paired (**b**) or unpaired (**i**) two-sided *t* test, one-way ANOVA and Dunnett's multiple comparisons test compared to vehicle (**d, e, j**) or IL-1β (**j**), two-way ANOVA and Šidák's multiple comparisons test (**f, g**), or Mann-Whitney test (**h**). Data are represented as individual measurements and mean ± SEM, box-and-whisker plots (line inside box = median; box limits = first and third quartiles; whisker ends = minima and maxima), or violin plots. Source data are provided as a Source Data File.

## The adipogenic stimulation by IL-1β is limited to the early differentiation stage

To get further insights on the function of IL-1β in adipocyte differentiation, we treated hASCs with IL-1β during differentiation and performed RNA-seq analysis at different time points. The number of differentially expressed genes in IL-1β-treated cells compared to vehicle diminished from early to late differentiation stage (Fig. 5a). Genes related to adipogenesis and lipid handling were upregulated by IL-1β on day 2 and day 5, but not day 9, while inflammatory pathways were upregulated at all time points (Fig. 5b), suggesting that only short and/or early IL-1β exposure promotes adipogenesis/maturation. In line with this, prolonging the IL-1β treatment until full differentiation of

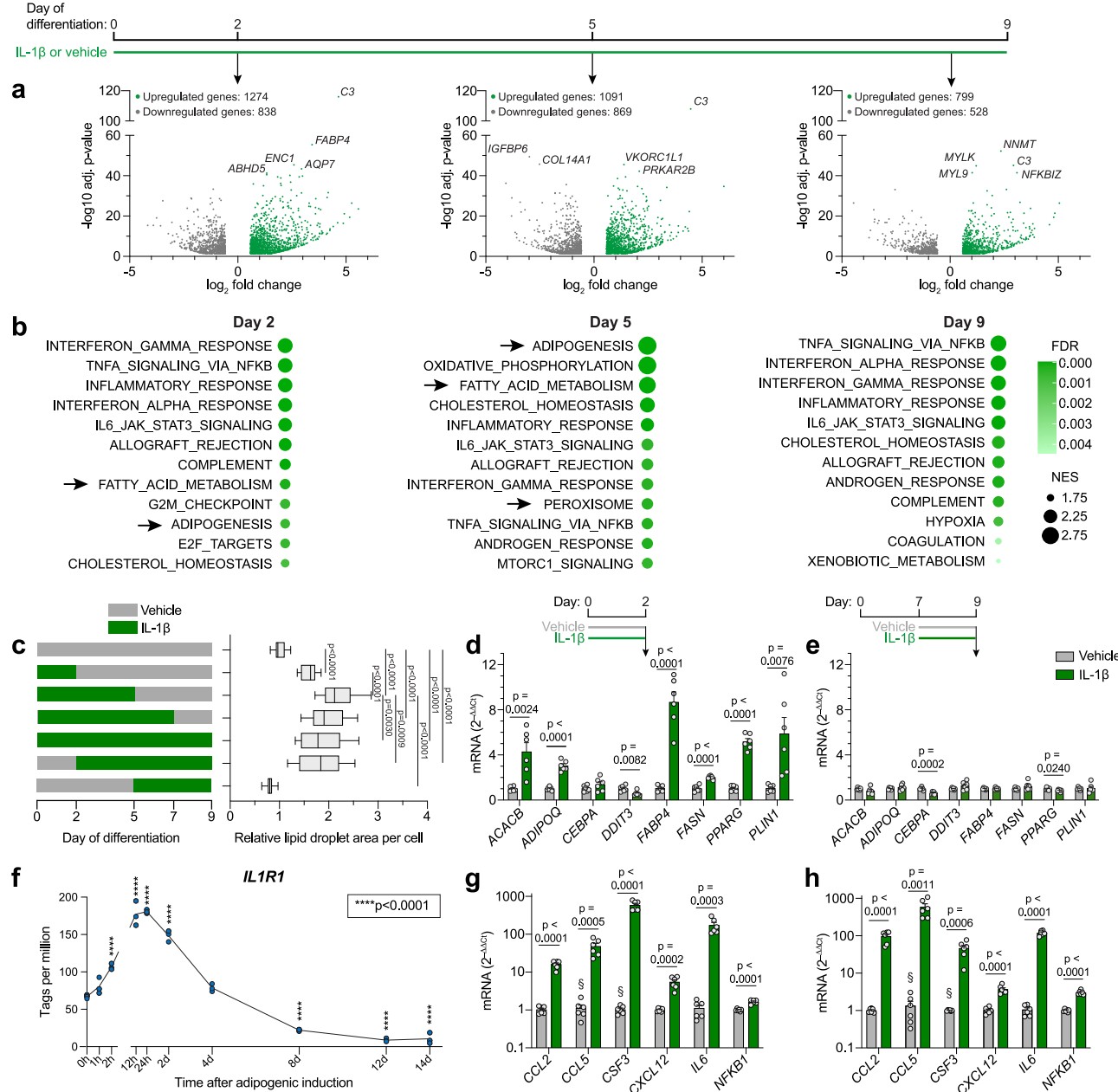

**Fig. 5 | The adipogenic stimulation by IL-1β is limited to the early differentiation stage.** **a** RNA-seq of differentiating hASCs treated with IL-1β from start of adipogenic induction and collected on day 2, 5, and 9 of differentiation. Volcano plots show genes significantly regulated by IL-1β (adjusted *p*-value < 0.05, fold change > 1.5 and < −1.5) compared to vehicle, as analyzed by Wald test corrected for multiple testing using the Benjamini-Hochberg method. The five most significantly regulated genes on each day are labelled (n = 4 independent experiments). **b** Gene Set Enrichment Analysis of RNA-seq data described in (**a**). The 12 most highly enriched Hallmark gene sets (by normalized enrichment score, NES) in IL-1β-treated cells on each day are shown. Arrows indicate pathways directly related to adipogenesis or lipid handling. **c** Lipid droplet accumulation on day 9 of differentiation in hASCs treated with IL-1β at indicated differentiation stages (n = 4 independent experiments with ≥ 4 replicates (52 datapoints for vehicle and 16 for the rest)). **d, e** Adipogenesis- and

adipocyte phenotype-related gene expression in hASCs treated with IL-1β on day 0 (**d**) or 7 (**e**) of differentiation and collected 48 h later (n = 3 independent experiments with 2 replicates). **f** *IL1R1* expression, measured by Cap Analysis of Gene Expression (CAGE), of hASCs at indicated time points after adipogenic induction (n = 3 replicates). **g, h** Expression of pro-inflammatory genes in hASCs treated with IL-1β on day 0 (**g**) or 7 (**h**) of differentiation and collected 48 h later (n = 3 independent experiments with 2 replicates). Statistical analyses by one-way ANOVA and Dunnett's multiple comparisons tests compared to vehicle (**c**), to IL-1β treatment days 0–5 (**c**), or to 0 h (**f**), or two-sided unpaired *t*-test (**d, e, g, h**). IL-1β: 10 ng/ml in all panels. Data are represented as box-and-whisker plots (line inside box = median; box limits = first and third quartiles; whisker ends = minima and maxima) or individual measurements and mean ± SEM. § denotes conditions with undetectable mRNA levels in all or some samples. Source data are provided as a Source Data File.

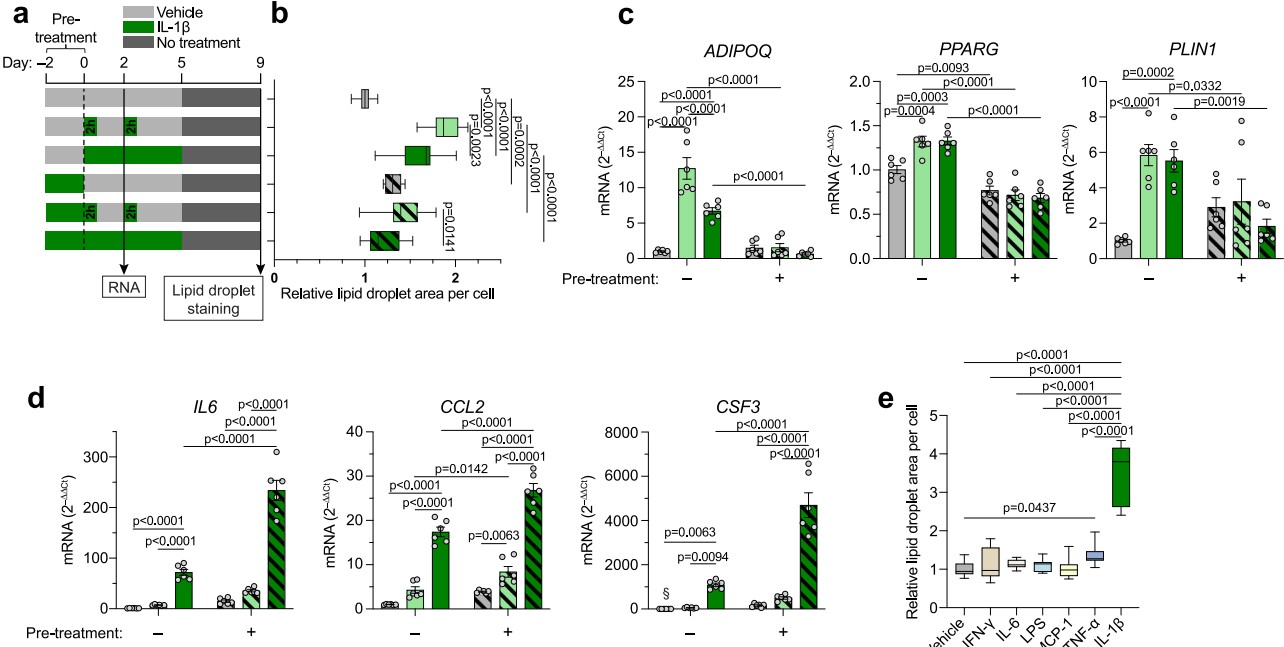

**Fig. 6 | Distinct effects of acute and chronic IL-1β exposure on adipogenic and inflammatory response. a** Schematic of treatments in experiments shown in (**b**–**d**). **b** Lipid droplet accumulation in hASCs treated as shown in (**a**) (n = 3 independent experiments with 4 replicates (12 datapoints per condition). **c, d** Expression of adipogenic (**c**) and pro-inflammatory (**d**) genes in hASCs treated as shown in (**a**). Conditions are color/pattern coded as in (**b**) (n = 3 independent experiments with 2 replicates). **e** Lipid droplet accumulation on day 9 of differentiation in hASCs exposed to IFN-γ (10 ng/ml), IL-6 (10 ng/ml), LPS (100 ng/ml),

MCP-1 (20 ng/ml), TNF-α (10 ng/ml), IL-1β, or vehicle for 2 h on day 0 and day 2 (n = 3 independent experiments with ≥ 4 replicates (14 datapoints for vehicle and 12 for the rest)). Statistical analyses by two-way ANOVA and Tukey's and Šidák's multiple comparisons tests (**b**–**d**) or one-way ANOVA and Dunnett's multiple comparisons tests compared to vehicle or to IL-1β (**e**). IL-1β: 10 ng/ml in all panels. Data are represented as box-and-whisker plots (line inside box = median; box limits = first and third quartiles; whisker ends = minima and maxima) or individual measurements and mean ± SEM. Source data are provided as a Source Data File.

hASCs at day 13 did not affect lipid accumulation or cell number and resulted in a downregulation of *PPARG* and *ADIPOQ* expression (Supplementary Fig. 6a–c). Therefore, to determine whether the adipogenic effect of IL-1β is differentiation stage-specific, we next limited IL-1β exposure to specific periods of differentiation. On day 9, lipid droplet accumulation was highest in cells treated from day 0 to day 5 (Fig. 5c). Continuing treatment past day 5 resulted in no additional, or even a slightly diminished formation of lipid droplets, and starting treatment at day 5 had no effect at all (Fig. 5c). Further, in a separate series of experiments where cells in distinct differentiation stages were treated with IL-1β for the last 48 h prior to collection, a gene expression pattern indicative of stimulated adipogenesis (including downregulation of the negative adipogenic regulator *DDIT3*[49]) was observed in early- (day 2) (Fig. 5d), but not late-stage (day 9) (Fig. 5e) cells. In line with this, we noted a rapid upregulation of *IL1R1* expression in hASCs upon adipogenic induction, which started to decline already after one day, and reached very low levels at the end of differentiation (Fig. 5f), substantiating the low *IL1R1* expression we observed in mature adipocytes (Fig. 3e). Interestingly, IL-1β induced expression of inflammatory genes independent of differentiation stage (Fig. 5g, h). Taken together, these results show that the adipogenic, but not inflammatory effects of IL-1β are limited to cells in an early stage of differentiation.

### Distinct effects of acute and chronic IL-1β exposure on adipogenic and inflammatory response

We next modelled an environment where IL-1β levels are increased either transiently (as observed postprandially[26]) or chronically (as observed in obesity) (Fig. 6a). In the first case, hASCs were exposed only to short (2 h) pulses of IL-1β from the start of adipogenic induction (Fig. 6a). Compared to continuous treatment, transient exposure induced greater lipid accumulation and *ADIPOQ* expression, whereas there was no difference in the increase of *PPARG* and *PLIN1*

transcription (Fig. 6b, c). In the second, "obese-like" case, cells were treated with IL-1β for two days (pre-treatment) prior to adipogenic induction (Fig. 6a). Pre-treatment blocked the stimulatory effects of IL-1β on lipid accumulation (Fig. 6b) and adipogenic gene expression (Fig. 6c). Interestingly, induction of pro-inflammatory cytokines was regulated in the opposite manner. IL-1β-induced expression of *IL6*, *CCL2*, and *CSF3* was markedly augmented by pre-treatment as well as by continuous exposure (Fig. 6d). This suggests that the IL-1β resistance induced by pre-treatment is specific to its pro-adipogenic signals and is not a general desensitization to the cytokine. Importantly, *IL1R1* expression was not decreased by pre-treatment or continuous exposure (Supplementary Fig. 7).

Since transient IL-1β exposure strongly promoted differentiation, we investigated whether general acute inflammatory signals exert pro-adipogenic effects. In line with the continuous exposure experiments (Fig. 4j), only IL-1β potently promoted lipid accumulation in the acute setting (Fig. 6e). Overall, these findings show that a transient IL-1β burst is sufficient to promote adipogenesis and that chronically elevated levels desensitize the progenitors to these potentially beneficial effects of IL-1β, and instead aggravate tissue inflammation.

### IL-1β increases C/EBPδ and C/EBPβ expression and their subsequent binding near adipogenic genes

To investigate which adipogenic signals are induced by IL-1β, we examined its effect on hASC differentiation in suboptimal differentiation media, lacking individual adipogenic factors. IL-1β treatment completely restored the loss of differentiation capacity caused by the absence of 3-isobutyl-1-methylxanthine (IBMX), but could not substitute insulin, rosiglitazone or dexamethasone (Fig. 7a). Similarly, IL-1β could completely or partially replace IBMX in the induction of adipogenic genes (Fig. 7b). This suggests that IL-1β and IBMX employ similar signal transduction pathways. A well-known role of IBMX in

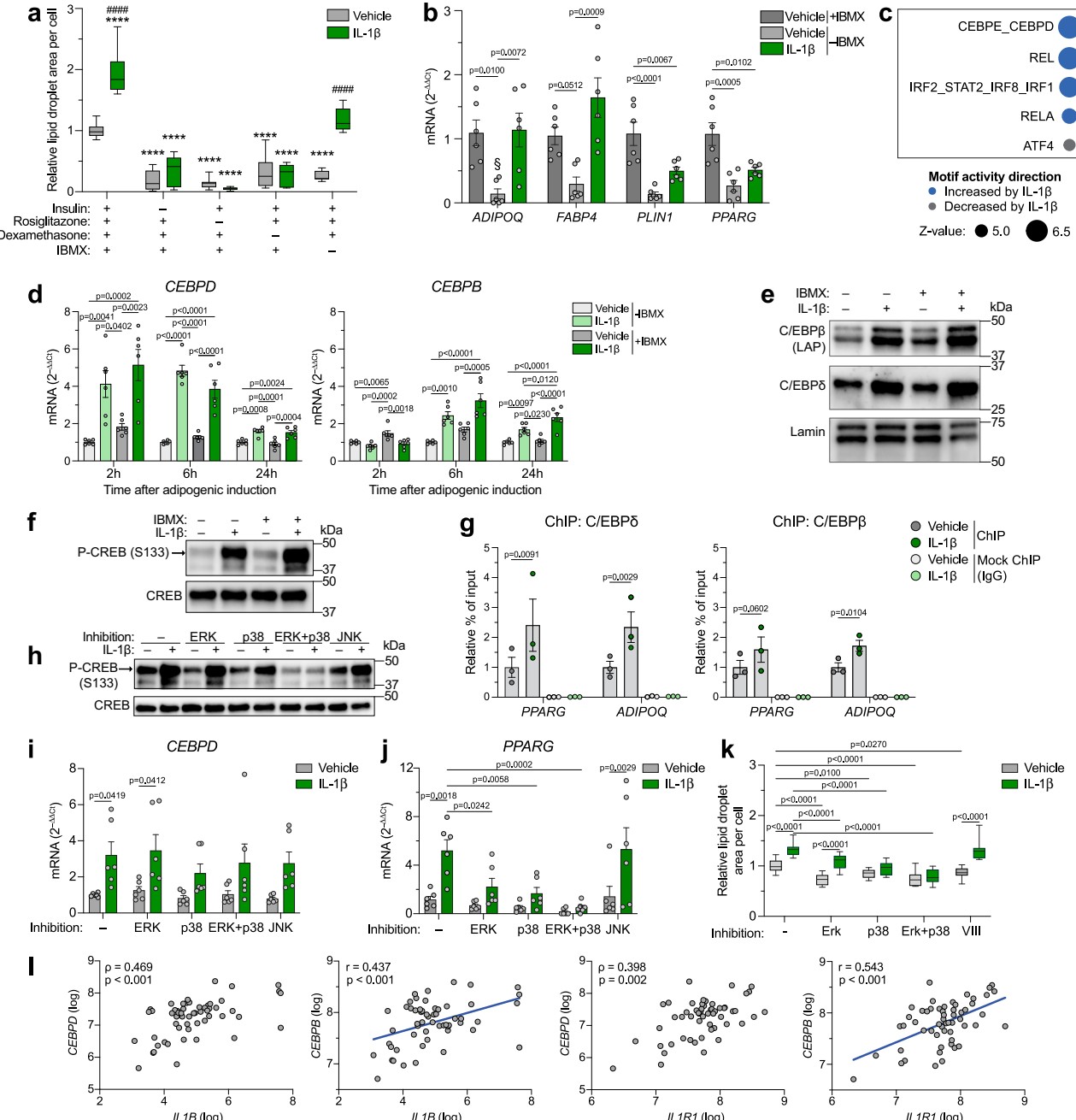

**Fig. 7 | IL-1β increases expression of C/EBPδ and C/EBPβ and their subsequent binding near adipogenic genes. a** Lipid droplet accumulation in hASCs differentiated for 9 days in complete (n = 18 datapoints) or suboptimal (n = 12 datapoints) adipogenic cocktail and +/− IL-1β for the first five days (n = 3 independent experiments with ≥ 4 replicates). **b** Adipogenic gene expression in hASCs 2 days after adipogenic induction +/− IBMX and IL-1β (n = 3 independent experiments with 2 replicates). § denotes conditions with undetectable mRNA levels in all/some samples. **c** Top 5 predicted IL-1β-regulated motifs in hASCs 2 days after adipogenic induction +/− IL-1β, analyzed by ISMARA (n = 4 independent experiments). **d** Gene expression at indicated time points after adipogenic induction +/− IBMX and IL-1β (n = 3 independent experiments with 2 replicates). **e, f** Western blot on protein collected 24 h (**e**) or 30 min (**f**) after adipogenic induction +/− IBMX and IL-1β (one representative blot from 3 independent experiments). **g** ChIP-qPCR of hASCs 20 h after adipogenic induction +/− IL-1β (n = 3 independent experiments). Data shown as % of input normalized to vehicle in the C/EBPδ- or C/EBPβ-pulldown. **h** Western blot on protein collected 30 min after adipogenic induction with IL-1β and indicated

inhibitor(s) (one representative blot from 3 independent experiments). **i–k** Gene expression (**i, j**) and lipid droplet accumulation (**k**) in hASCs after 2 h (**i**), 24 h (**j**) and 9 days (**k**) of differentiation +/− IL-1β for the first 2 h and indicated inhibitor(s) for the first 2 (**i**) or 24 (**j, k**) h (n = 3 independent experiments with 2 (**i, j**) or ≥ 4 (**k**) replicates (**k**: 18 datapoints for inhibitor-free conditions and 12 for the rest)). **l** Pearson (r coefficient) or Spearman (ρ coefficient) correlations between scWAT mRNA levels of *IL1B/IL1R1* and *CEBPD/CEBPB* in a clinical cohort (n = 56). Two-tailed p-values. IL-1β: 10 ng/ml in all panels. Statistical analyses by two-way ANOVA and Dunnett's (****p < 0.0001 in **a**) or Šidák's (####p < 0.0001 in **a**) multiple comparisons test against vehicle of complete adipogenic cocktail or within each adipogenic cocktail condition, respectively (**a**), and against matched treatment (vehicle or IL-1β) without inhibitor or within each inhibitor condition, respectively (**i-k**), one-way ANOVA and Tukey's multiple comparisons test (**b, d**), or two-sided ratio paired t test (**g**). Data are represented as box-and-whisker plots (line inside box = median; box limits = first and third quartiles; whisker ends = minima and maxima) or individual measurements and mean ± SEM. Source data are provided as a Source Data File.

adipogenesis is its upregulation of C/EBPβ[12,50,51] via activation of the transcription factor CREB[52], which has also been shown to bind to the promoter region of C/EBPδ[50]. C/EBPβ and C/EBPδ are two closely linked adipogenic transcription factors well known for their crucial role during early differentiation[53–55]. An integrated motif activity response analysis (ISMARA) of our RNA-seq data from differentiating IL-1β-treated hASCs collected on day 2 predicted the C/EBPδ and C/EBPε motifs to be the most significantly activated by IL-1β (Fig. 7c). Since only CEBPD, but not CEBPE mRNA was detected in this RNA-seq data set (see data deposited at NCBI gene expression omnibus, accession number GSE237151), changes in expression of genes associated to these transcription factors were likely solely driven by C/EBPδ. Taken together, these observations suggested C/EBPβ and/or C/EBPδ as mediators of the adipogenic effect of IL-1β. In line with this, IL-1β increased gene expression of both transcription factors during the first day of differentiation, both in the presence and absence of IBMX in the differentiation media, with the induction of CEBPD being stronger and more rapid compared to that of CEBPB (Fig. 7d), suggesting the former may be mediated directly via CREB and that the latter is an indirect effect. Interestingly, IL-1β upregulated these transcription factors, especially CEBPD, more potently than IBMX. The same trend was observed at the protein level, where IL-1β strongly increased C/EBPδ and the two isoforms of C/EBPβ containing the transactivation-domain (liver activating protein, LAP and LAP*) both in the absence and presence of IBMX, and the expression induced by IL-1β was stronger than that by IBMX (Fig. 7e). This suggests that the ability of IL-1β to substitute IBMX during differentiation (Fig. 7a) is due to its IBMX-independent upregulation of C/EBPδ and C/EBPβ. IL-1β treatment also led to a rapid (30 min) increase in CREB-phosphorylation to a greater extent than IBMX (Fig. 7f), suggesting that IL-1β induces CEBPD expression (and possibly CEBPB indirectly) through activation of CREB. Importantly, chromatin immunoprecipitation (ChIP)-qPCR showed an enrichment of C/EBPδ and C/EBPβ at known C/EBP-binding sites in the promoter regions of PPARG and ADIPOQ by IL-1β (Fig. 7g), suggesting that the IL-1β-induced expression of these transcription factors is also coupled to their increased occupancy near adipogenic genes.

IL-1β is a known activator of the mitogen-activated protein kinases (MAPKs) extracellular signal-regulated kinase (ERK), p38 and c-Jun N-terminal kinase (JNK)[56], of which the two former pathways can activate CREB[57]. Therefore, to elucidate through which signaling pathway IL-1β activates CREB and upregulates CEBPD, we used pharmacological inhibitors to block each of these pathways during the first day of differentiation. ERK and p38 were also blocked simultaneously to avoid possible compensatory effects between these pathways on CREB activation. ERK or JNK inhibition alone did not affect the ability of IL-1β to activate CREB (Fig. 7h). Conversely, IL-1β-induced CREB phosphorylation was partially attenuated by p38 inhibition and completely abolished by simultaneous inhibition of p38 and ERK (Fig. 7h), suggesting that IL-1β mainly activates CREB via p38 but that loss of this signaling pathway may be somewhat compensated for by ERK, but not vice versa. Similarly, the p38 inhibitor, both alone and in concert with ERK inhibition, also attenuated the IL-1β-induced upregulation of CEBPD (Fig. 7i), supporting a role of CREB in this transactivation. Importantly, inhibiting p38, either alone or together with ERK, also attenuated IL-1β-induced PPARG upregulation (Fig. 7j) and completely abolished IL-1β-stimulated lipid droplet accumulation (Fig. 7k), supporting the involvement of p38-mediated CREB activation and subsequent CEBPD upregulation for the adipogenic effect of IL-1β. Conversely, ERK inhibition alone blocked PPARG upregulation by IL-1β (Fig. 7j) but failed to block the increase in CEBPD expression (Fig. 7i) and lipid droplet accumulation (Fig. 7k), whereas JNK inhibition attenuated IL-1β-stimulated CEBPD upregulation (Fig. 7i) but not PPARG upregulation (Fig. 7j) or lipid droplet accumulation (Fig. 7k). Taken together, these results

indicate that IL-1β promotes adipogenesis through the p38 signaling pathway, likely partly via p38-mediated activation of CREB and subsequent upregulation of CEBPD.

In line with our in vitro findings, scWAT expression of IL1B and IL1R1 correlated positively to that of both CEBPD and CEBPB in a clinical cohort of women with and without obesity[46] (Fig. 7l). No association between CEBPD or CEBPB expression and BMI (CEBPD: Spearman's ρ = 0.011, p = 0.933; CEBPB: Spearman's ρ = 0.075, p = 0.584) or body fat percent (CEBPD: Spearman's ρ = 0.002, p = 0.988; CEBPB: Spearman's ρ = 0.051, p = 0.711) were observed. Further, analysis of publicly available human WAT microarray data[25] revealed a marked postprandial increase in expression of IL1B (Supplementary Fig. 8a), in line with findings in mice[26]. Interestingly, mRNA levels of CEBPD and CEBPB were also increased (Supplementary Fig. 8b, c), and the postprandial change in IL1B expression correlated positively to the change in CEBPD and CEBPB expression (Supplementary Fig. 8d, e). This suggests that our in vitro observations reflect a physiological function and supports the hypothesis that postprandial surges in IL-1β may promote adipogenesis through upregulation of C/EBPδ and C/EBPβ in human WAT.

## Discussion

In the present study, we investigated the role of IL-1β in WAT remodeling and report that this cytokine promotes adipocyte differentiation by directly targeting adipose-derived stem cells. Deletion of IL-1R1 in mature adipocytes had no metabolic consequences, whereas whole-body IL1R1-KO mice displayed reduced body weight gain, WAT mass, and adipocyte size. Based on IL1R1 expression pattern among WAT-resident cells, adipocyte precursors were identified as the main target of IL-1β in both human and murine WAT. Thus, loss of IL-1 signaling in this cell population likely contributed to the phenotype of the IL1R1-KO mice, possibly reflecting an impaired fat mass development due to an inability of adipocyte progenitors to properly differentiate and fully mature, similarly to the observation of very small adipocytes in insulin resistant individuals[58]. Importantly, direct measurement of new adipocyte formation in vivo by EdU tracing revealed a substantially decreased production of new fat cells in both gWAT and scWAT of IL1R1-KO mice upon HFD-feeding. The direct effect of IL-1β in promoting adipocyte differentiation was robustly validated in three different in vitro models, and further supported by human clinical data showing a strong link between reduced WAT IL-1 signaling and hypertrophic WAT. Based on our findings, we propose that IL-1β acts to direct energy storage towards newly formed adipocytes through direct stimulation of adipogenesis, suggesting a critical physiological role in healthy WAT expansion as a response to increased food intake.

C57BL/6 J mice, a strain associated to redox imbalances and metabolic alterations[59–62], were used to investigate WAT morphology and adipogenesis in knockout animals, while WT C57BL/6 N mice were employed for pharmacological IL-1 blockage. Despite recognized genetic and metabolic differences between the two sub-strains, we obtained robust results by using a large number of mice in three independent cohorts with age-, sex-, commercial vendor-, and genetic background-matched controls. In contrast to our results in IL1R1-KO mice, IL-1Ra treatment in WT mice increased gonadal adipocyte formation. Although we cannot fully exclude the effect of genetic background, this discrepancy might reflect a true difference between transient pharmacological IL-1 blockage in adulthood and an inborn, chronic lack of IL-1 signaling, which may warrant further investigations.

In certain aspects, our observations contrast published studies. Recent publications showed that adipocyte-specific deletion of adaptor molecules (Myd88, IRAK2, and IRAKM) downstream of IL-1 signaling led to metabolic alterations in HFD-fed mice[63,64]. It is worth noting that these mouse models also block signal transduction of toll-like receptors (TLRs), suggesting that the observed metabolic changes in these KO mice are, in some extent, mediated by the disruption of TLR signaling in adipocytes. In this current study, we observed reduced

body weight, fat mass, and adipocyte size upon genetic or pharmacological blockade of IL-1 signaling, while previous publications reported increased or unaltered body weight and fat mass, and decreased, unaltered or augmented adipocyte size of IL1R1-KO mice[65–69]. These discrepancies could be explained by differences in diet, genetic background, housing conditions, use or not-use of littermate controls, age, and sex of the mice. Further, IL-1R1 ligands (IL-1α and IL-1β) may exert distinct biological effects, as IL-1β-deficient mice show increased WAT mass[70], while IL-1α-KO mice have a remarkable decreased WAT mass and adipocyte size[71]. Overall, these varying results highlight an important but complex role of the IL-1 system in energy homeostasis, which remains to be fully elucidated.

The exclusivity of the adipogenic stimulation by IL-1β to early-stage progenitors explains the lack of phenotype displayed by IL1R1[AKO] mice. This conditional mouse model relies on adiponectin expression to delete *Il1r1* sequences, and *Adipoq* is not expressed during the initial differentiation period. The mediators of the adipogenic effect of IL-1β, C/EBPδ and C/EBPβ, are only important for adipogenesis during the early stage of differentiation[12,13,53]. In fact, while overexpression of C/EBPβ LAP in hASCs promotes *PPARG* and *FABP4* expression at the start of adipogenic induction, it reduces their expression, and that of several other adipogenic genes, during the middle and late stages of differentiation[72]. Since IL-1β exposure induced inflammation in our late-stage cells, the inability of IL-1β to stimulate adipogenesis/lipid accumulation during late differentiation was not caused by a general unresponsiveness to the cytokine, but likely due to the inability of C/EBPδ and C/EBPβ to promote adipocyte maturation in late-stage cells.

Acute exposure (2 h) to IL-1β promoted adipocyte differentiation to a larger degree than continuous treatment, suggesting that the transient postprandial surge in IL-1β[26,37] is sufficient to relay its adipogenic signals. This also implies that prolonged IL-1R1 signaling ultimately suppresses the adipogenic program that it initially stimulated. Considering the stronger inflammatory gene induction by continuous IL-1β exposure compared to acute, and the well-established anti-adipogenic properties of inflammatory signals[73–76], inflammation likely contributed to this adipogenic suppression. For instance, the central inflammatory transcription factor nuclear factor κ B (NFκB) can redistribute transcriptional co-factors from enhancers of genes related to adipocyte metabolism to those of inflammatory genes[77]. Stopping IL-1β treatment after 2 h likely reduces NFκB activity, and thereby its adipogenic inhibition. Similarly, inflammation-mediated suppression of differentiation likely contributes to the desensitization of IL-1β-induced adipogenesis in pre-treated hASCs, as these cells displayed an augmented inflammatory induction. This suggests that progenitors are unable to respond to the physiological adipogenic effects of IL-1β in the obese WAT due to chronically activated inflammatory pathways. In line with this, progenitor subpopulations with a pro-inflammatory profile are increased in the WAT of mice with diet-induced obesity[78,79].

The distinct adipogenic and inflammatory effects of IL-1β on hASCs depending on treatment timing and duration may explain why some previous studies described an anti-adipogenic (and insulin resistance-inducing) effect of IL-1β[35,80,81]. These observations were made after continuous treatment with IL-1β. Thus, any potential promotion of adipogenesis during the early stage may have been counteracted by inhibitory effects during the later stage of differentiation and/or by prolonged treatment. In line with this, we also observed that the IL-1β-induced promotion of lipid accumulation was lost in fully matured cells treated throughout the whole 13-day differentiation period, concomitant with downregulation of *PPARG* and *ADIPOQ* expression.

IL-1β was able to completely replace IBMX as an adipogenic stimulator. IBMX is a broad-spectrum phosphodiesterase inhibitor, which increases cAMP levels, thereby inducing *CEBPB* expression[52] via activation of CREB[82]. In addition, CREB has been shown to bind to CRE

elements in the *CEBPD* promoter[50]. C/EBPδ and C/EBPβ are two closely linked pro-adipogenic transcription factors that can form heterodimers, which bind to the same promoters as their respective homodimers[83]. The IL-1β-induced upregulation of *CEBPD* (≤ 2 h) was faster than that of *CEBPB* (> 2 h, ≤ 6 h), suggesting that the latter is indirect. We observed a rapid and potent activation of CREB by IL-1β, which was mediated mainly via p38 signaling and likely, at least partly, responsible for the induction of *CEBPD* expression. In line with our results, a recent study showed a pro-adipogenic effect of IL-1β in dermal fibroblasts, which was partly attributed to p38- and NFκB-mediated CREB-activation, with subsequent induction and activation of *CEBPB*[84]. Additionally, IL-1β was also shown to inhibit the adipogenesis-suppressing Wnt-β catenin pathway[84]. Using ChIP-qPCR, we confirmed that IL-1β increased the binding of both C/EBPδ and C/EBPβ to DNA regions near the *PPARG* and *ADIPOQ* genes when the mRNA levels of the latter gene are still very low. During early adipogenesis, C/EBPβ and C/EBPδ have been shown to mark DNA regions that later are bound by PPARγ and C/EBPα[15]. Thus, C/EBPδ and C/EBPβ likely mediate the adipogenic effect of IL-1β by both directly upregulating *PPARG* expression, and indirectly promoting expression of PPARγ- and C/EBPα-target genes by recruiting these adipogenic master regulators to relevant genomic sites.

Some years ago, the concept of healthy inflammation in adipose tissue emerged. Philipp Scherer's group elegantly showed that inflammatory pathways within adipocytes play a crucial role in facilitating adipose tissue expansion. The suppression of pro-inflammatory pathways in adipocytes led to adipose tissue dysfunction, ectopic lipid deposition, impaired glucose metabolism, and systemic inflammation. Moreover, the induction of acute inflammation in intact adipose tissue stimulated adipogenesis[19,21]. Our results complement the work of Zhu and Wernstedt-Asterholm et al., identifying IL-1β as a key modulator of healthy inflammation as a modulator of adipose tissue expansion.

In conclusion, we show that IL-1β regulates WAT remodeling by directly promoting differentiation of adipocyte precursors. This can be induced by a transient IL-1β burst, as observed postprandially, and probably has a physiological role in allowing adipose tissue to adapt during caloric excess. However, chronically elevated levels of IL-1β, as seen in obesity, desensitize progenitor cells to these potentially beneficial effects of IL-1β, and instead exacerbate inflammation. These results have relevant implications for the understanding of the distinct roles of WAT inflammation in health and obesity-related metabolic diseases, paving the way for the development of new IL-1 modulation strategies for the treatment of obesity-related metabolic diseases.

## Methods
### Mouse models
All animal experiments were performed according to the Swiss veterinary law and were approved by Swiss authorities (cantonal veterinary office of Basel). Adipocyte-specific deletion of IL1R1 (IL1R1[AKO]) was achieved by breeding homozygous IL1R1 floxed mice (IL1R1[FF], JAX: 028398) with adiponectin-Cre transgenic mice (JAX: 010803, obtained from Prof. Dr. Daniel Konrad (Universitäts-Kinderspital Zürich)). Littermate mice (IL1R1[FF]) were used as control. Male and female mice were used, as described in the figure legend. WT male mice on a C57BL/6 J genetic background (JAX: 000664) were used as control of IL1R1-deficient male mice (IL1R1-KO, JAX: 003245). Prior to intercrossing, all animals were fully backcrossed (at least 10 generations) onto a C57BL/6 J genetic background. WT male mice on a C57BL/6 N genetic background were used during the experiments with IL-1Ra. Body weight was measured weekly. To increase readability, not all body weight data are shown in the figures. Daily food intake was measured by the food given/left in the rack feeder of the cage. Mice received standard chow (#3436, Kliba Nafag, Kaiseraugst, Switzerland) or lard-based 60 kJ% HFD (#E15742-34, ssniff Spezialdiäten GmbH, Soest, Germany) and water ad libitum. Eight to twelve-week-old male

mice were used, according to the experimental design. Eight to twelve-week-old mice were HFD-fed for 4 days, 6 weeks, 9 weeks, or 35 weeks, according to the experimental design. Mice were housed in groups of 2–6 mice in individually ventilated cages (Indulab, Gams, Switzerland) with standard bedding, nesting material, and a house. The animal facility was pathogen-free, temperature- ($21 \pm 2\,°C$) and humidity-controlled (50–60%) with a 12 h dark/12 h light cycle (lights on at 6:00), at the University of Basel. Mouse euthanasia was performed by trained researchers, using a chamber with gradual carbon dioxide exposure, followed by cardiac puncture.

## Adipocyte and stromal vascular fraction isolation from murine and human WAT

Human stromal vascular fraction as well as murine stromal vascular fraction used for cell culture after magnetic bead isolation (see below) was isolated from WAT[85] by incubating minced WAT pieces in roughly equal volume of KRP buffer with 0.05% collagenase (#C0130, Sigma-Aldrich, St Louis, USA) and 4% BSA (#A4503, Sigma-Aldrich) at $37\,°C$ for 40–90 min with manual shaking every 10 min, followed by filtration through a $200\,\mu m$ nylon mesh and then leaving it for a few minutes to allow the oil and mature adipocytes to float to the top. After collection of the stromal vascular fraction, the mature adipocytes were washed with PBS and the infranatant solution was again collected and added to the stromal vascular fraction, which was then centrifuged, resuspended in PBS, and centrifuged again. The cells were either used immediately or resuspended in fetal bovine serum (FBS) with 10% DMSO and stored in liquid nitrogen. Isolation of all other murine stromal vascular fraction and adipocytes[86] was performed as follows. WAT was harvested, minced, and placed in 4% BSA (#A6003, Sigma-Aldrich) solution in Krebs-Ringer HEPES buffer (KRH). Tissue digestion was obtained after 40–45 min incubation with 1.5–2.9 mg/mL collagenase type I (#LS004197, Worthington, Freehold, USA) $37\,°C$ and 400–450 rpm (50 mL ThermoMixer, Eppendorf). Of note: the digestion of scWAT from IL1Ra-treated mice was harder, requesting elevated collagenase concentration. The digested WAT was filtered through a nylon mesh (pluriStrainer® $200\,\mu m$, pluriSelect, Germany) and washed three times with 1% BSA-KRH. After each wash, the cell suspension rested at room temperature for 6 min to separate the floating adipocytes from infranatant. The infranant solution, containing the stromal cells, was removed and kept on ice. Packed adipocytes were immediately used in downstream processing, or frozen in liquid nitrogen and stored at $-80\,°C$. Stromal vascular fraction was washed twice, resuspended in freezing media (fetal bovine serum with 10% DMSO), and stored in liquid nitrogen prior to downstream processing.

## Histological analysis

Tissues were dissected, weighed, and fixed in 4% formaldehyde for 24 h at $4\,°C$. Paraffin-embedding tissues were sectioned into $10\,\mu m$ slices (HM340E microtome, Thermo Fisher Scientific, Waltham, USA) and subjected to H&E staining (Gemini AS, Epredia, New Hampshire, USA). The stained slides were imaged with a light microscope (Nikon ECLIPSE Ti2, Nikon, Switzerland) and adipocyte area was determined by Adiposoft (Fiji, ImageJ, USA). In brief, 10X objective was used to acquire representative images ($0.72\,\mu m$/pixel) of WAT. Adipocytes at the border of the image frame were automatically excluded and the diameter threshold was set as 20–200 $\mu m$[87]. The adipocyte area frequency distribution was calculated in 500 increments from 0 to 15000 $\mu m^2$.

## In vivo glucose uptake assay

12-week-old IL1R1[AKO] and littermates, HFD-fed for 4 days, were injected with $1\,\mu g$/kg body weight IL-1β (#401-ML, RD System), or 1 U/kg body weight insulin (Actrapid HM, NovoNordisk, Basvaerd, Denmark), or saline 20 min prior to an injection containing nonmetabolizable 350 $\mu Ci$/kg $^3H$-2-deoxy-glucose (#NET328A001MC, PerkinElmer,

Switzerland). 50 min after the first injection, mice were euthanised for organ harvesting. Adipose tissue was snap-frozen in 1 mL lysis buffer (10% glycerol, 5 mmol/L 2-mercaptoethanol, 2.3% SDS, 62.5 mmol/L Tris, 6 mol/L urea, 1 $\mu mol$/L NaF, phosphatase inhibitor cocktail (#P5726, Sigma)) and stored at $-80\,°C$ for later measurement of $^3H$-2-deoxy-glucose-6-phosphate[88]. Alternatively, WT male mice were simultaneously injected with the $^3H$-2-deoxy-glucose (350 $\mu Ci$/kg) and IL-1β ($1\,\mu g$/kg body weight) or saline 18 min prior to euthanasia for organ harvesting and later measurement of glucose uptake in adipose tissue and isolated adipocytes. Adipose tissue was disrupted using a tissue lyser, at $4\,°C$. The lysate was incubated at $95\,°C$ for 5 min, followed by centrifugation at 12000 $g$ for 15 min. The supernatant was 10-fold diluted in distilled water and purified by column chromatography (#7316212, Bio-Rad). The column was prepared with 10 mL of 1.25 mol/L NaOH, followed by 5 mL of distilled water, 5 mL of 1 mol/L acetic acid, and 5 mL distilled water. 2 mL sample was loaded on the column and washed with 5 mL distilled water prior to the elution in 6 mL of elution buffer (0.2 mol/L formic acid, 0.6 mol/L ammonium acetate). Purified $^3H$-2-deoxy-glucose-6-phospate was 1:8 mixed with liquid scintillation cocktail (#6013326, Ultima Gold, PerkinElmer) and isotope counting performed in a liquid scintillation analyzer (Tri-Carb 1900TR, Packard).

## In vivo EdU incorporation

EdU was used as an indirect marker for differentiation of adipocyte progenitor cells. Proliferating progenitors, but not post-mitotic adipocytes, incorporate EdU into DNA, resulting in EdU-labelled adipocytes upon cell differentiation. Eight-week-old mice were used for all EdU experiments, maintained on chow diet or HFD for 9 weeks. IL1R1-KO and controls were injected with 100 $\mu g$ EdU every 2 days, for the first 7 days. WT mice were: injected with 100 $\mu g$ EdU every 2 days, for 7 days (day 1 to day 7); daily treated with 10 mg/kg body weight IL-1Ra (Anakinra, Kineret, Sobi, Germany) at 4–6 pm, for 14 days (day 1 to day 14); HFD-fed for 9 weeks (starting at day 3). 9 weeks after the beginning of EdU treatment, the mice were euthanized, gWAT and scWAT mass was measured, and mature adipocytes were isolated. Frozen packed adipocytes were thawed in lysis buffer A (3 mmol/L MgCl$_2$, 10 mmol/L KCl, 10 mmol/L NaCl, 10 mmol/L Tris, 0.05% Nonidet P-40; volume: 3x adipocyte volume) and incubated for 10 min at room temperature while shaking at 400 rpm. The cells were lysed using a syringe with 23 G needle (3 strokes) and vigorous shaking. Lysis buffer B (Lysis buffer A with 0.6 mol/L sucrose; volume: 3x adipocyte volume) was immediately added and mixed to the lysed cells[89]. Nuclei were pelleted by 10 min centrifugation (1000 $g$, $4\,°C$), fixed in 3.7% formaldehyde (15 min at room temperature, shaking at 300 rpm), washed with 3% BSA, and permeabilized. EdU was stained with Alexa 488, followed by nuclear staining with Hoechst 33342, using the Click-iT™ Plus EdU Cell Proliferation Kit (#C10637, Invitrogen, Thermo Fisher Scientific) according to the manufacturer's instructions. Nuclei were analyzed by flow cytometry (BD LSR Fortessa, BD Biosciences, Franklin Lakes, USA). Data analyses were performed using FlowJo software (BD Biosciences) and expressed as the percentage of Hoechst-positive events that were also EdU-Alexa 488-positive (gating strategy is shown in Supplementary Fig. 9).

## Glucose and insulin tolerance tests

Prior to the intraperitoneal glucose tolerance test, mice were fasted for 6 h (from 8:00 to 14:00). Glucose was intraperitoneally injected at a dose of 2 g/kg of body weight and blood glucose was measured at the indicated times using a glucometer (Freestyle, Abbott Diabetes Care Inc., Switzerland). At time points 0, 15, and 30 min after glucose injection, blood was collected from the tail vein for determination of plasma insulin. Prior to the insulin tolerance test, mice were fasted for 3 h (from 10:00 to 13:00) followed by intraperitoneal injection of 1–2 U/kg body weight insulin (Actrapid HM, NovoNordisk). Plasma glucose was measured at the indicated time points.

## Clinical cohorts

A cohort of metabolically well-characterized women with (n = 30; mean age of 43 ± 2 SEM) and without (n = 26; mean age of 43 ± 3 SEM) obesity (all healthy)[46] was used to correlate expression of genes of the IL-1 system and clinical metabolic parameters. Transcriptomics data and mean adipocyte volume from these women were obtained from abdominal scWAT biopsies collected through needle aspiration. Human primary progenitor cells were derived from the scWAT of healthy women undergoing plastic surgery, where donors were anonymous and only age, sex and BMI were available to us. Both studies were approved by the regional ethics board of the Stockholm County Council, and all subjects signed written informed consents.

## Gene expression in human scWAT cell populations

Expression of *IL1B*, *IL1RN*, and *IL1R1* was analyzed in a microarray dataset[90] (GSE100795) of scWAT stromal vascular cell populations isolated by FACS and paired mature adipocytes from 3 individuals with and 3 without obesity. Stromal vascular fraction was stained with markers for progenitors (CD45⁻CD34⁺CD31⁻), M1 (CD45⁺CD14⁺CD206⁺CD11c⁺), and M2 (CD45⁺CD14⁺CD206⁺CD11c⁻) macrophages, as well as CD4⁺ (CD45⁺CD14⁻CD3⁺CD4⁺CD8⁻) and CD8⁺ (CD45⁺CD14⁻CD3⁺CD8⁺CD4⁻) T cells as described below for human progenitors and sorted using a FACSAria III equipped with 405, 488, 561 and 633 nm lasers and the Diva software (BD Biosciences). Total RNA was isolated with the RNeasy Micro Kit (Qiagen) according to the manufacturer's instructions. 10 ng of RNA from each cell population was amplified over four cycles and loaded onto Clariom™ D human microarray chips (Affymetrix) according to the manufacturer's instructions. Data were analyzed using Expression Console (Affymetrix) and the SST-RMA algorithm.

## Cell cultures

**Differentiation of freshly isolated hASCs.** Immediately after FACS-isolation, human primary progenitor cells derived from the scWAT of three healthy women (48-59 years old, BMI 29.4-32.1) undergoing cosmetic plastic surgery, were plated in growth medium, containing Dulbecco's modified eagle medium (DMEM)/F12 GlutaMAX (#31331-028, Gibco, Thermo Fisher Scientific) with 10% FBS, 100 U/ml Penicillin, and 100 μg/ml Streptomycin. The following day (day 0), adipogenesis was induced by adding differentiation medium, consisting of the growth medium without FBS, supplemented with 15 mmol/L HEPES buffer, 1.25 μg/ml fungizone, 100 nmol/L cortisol, 66 nmol/L insulin, 10 μg/ml transferrin, 33 μmol/L biotin, 17 μmol/L pantothenate, 1 nmol/L triiodothyronine, and 10 μmol/L rosiglitazone. Media was changed every 2–3 days. From day 4, differentiation medium without rosiglitazone was used. Cells were fixed on day 13.

**Differentiation of hASCs.** Passaged hASCs originating from the scWAT of a healthy male donor (16 years old, BMI 24) were used for most cell culture assays. These cells were isolated as described above for human stromal vascular fraction and expanded in proliferation media, consisting of low-glucose DMEM (#31885-023, Gibco) supplemented with 10% FBS, 0.01 mol/L HEPES buffer, 50 U/ml Penicillin, 50 μg/ml Streptomycin, and 2.5 ng/ml fibroblast growth factor 2 (FGF2)[91,92]. When cells reached confluence, FGF2 was removed from the media, and they were seeded to plates the next day. The following day (day 0), cells were washed twice with PBS, and differentiation was induced by adding differentiation media (one part Ham's F12 nutrient mix (#21765-029, Gibco) and one part proliferation media without FBS and FGF2, along with the following adipogenic stimuli: 860 nmol/L insulin, 10 μg/ml transferrin, 0.2 nmol/L triiodothyronine, 10 μmol/L rosiglitazone, 100 μmol/L IBMX, and 1 μmol/L dexamethasone). Media was changed every 2–3 days. Differentiation media without IBMX and dexamethasone was used from day 5.

**Differentiation of murine ASCs.** Immediately after isolation by magnetic beads, primary murine CD45-negative stromal vascular cells were plated in growth media, consisting of DMEM/F12 Glutamax (#31331-028, Gibco), with additional glucose added to 4.5 mg/ml, 10% donor calf serum, 50 U/ml Penicillin, and 50 μg/ml Streptomycin. The following day (day 0), adipogenesis was induced by adding differentiation media, consisting of the growth media with donor calf serum replaced by 10% FBS, 5 μg/ml bovine insulin, 0.25 μmol/L dexamethasone, 0.5 mmol/L IBMX, and 10 μmol/L rosiglitazone. On day 2, media was changed to differentiation media without dexamethasone, IBMX, and rosiglitazone. Cells were fixed on day 5.

## Cell culture treatments

Human recombinant IL-1β (#I9401 or #H6291), IL-6 (#SCU0001), TNF-α (#T6674), IFN-γ (#I17001), MCP-1 (#SRP3109), LPS (#L4524), IL-1Ra (#SRP3084) (all from Sigma-Aldrich), or vehicle (water and/or 0.0001% human albumin) was added to the cell culture media at indicated concentrations. IL-1Ra was added to the cell culture media 30 min prior to treatment with IL-1β. For experiments investigating transient ("postprandial-like") and chronic ("obese-like") treatments, hASCs were treated with vehicle or IL-1β (10 ng/ml) two days prior to adipogenic induction (pre-treatment). At start of adipogenic induction (day 0), cells were treated with IL-1β (10 ng/ml) (or vehicle). After 2 h, the media was changed to new differentiation media with IL-1β (10 ng/ml) (continuous treatment) or vehicle (transient treatment). This was repeated on day 2. From day 5, all cells received normal differentiation media without any treatments. RNA was collected on day 2 (prior to day 2 treatments) and lipid droplet accumulation was measured on day 9. In experiments using MAPK inhibitors, ERK, p38 or JNK were inhibited using U0126 (10 μmol/L: #9903; Cell Signaling), SB203580 (2.5 μmol/L; #S8307, Sigma-Aldrich) or JNK Inhibitor VIII (10 μmol/L; #420135, EMD Millipore), respectively. These inhibitors were added to the cell culture media 1 h prior to adipogenic induction and treatment with IL-1β.

## In vitro glucose uptake assay

In vitro glucose uptake was measured in differentiated hASCs by liquid scintillation counting using radiolabeled glucose. The day before the assay, insulin was removed from the differentiation media. The next day, cells were washed twice with glucose-free DMEM (#11966-025, Gibco) and then incubated for 3 h in the DMEM with 1 μmol/L glucose added. Media was then changed to DMEM with 1 μmol/L unlabeled glucose, 0.8 μCi/ml D-[3-³H]-glucose (#NET331A001MC, PerkinElmer), and IL-1β (or vehicle), and incubated for 2 h. After 3 washes with PBS, cells were lysed and scraped in 0.1% SDS in $H_2O$. A small volume of the lysate was saved for protein concentration measurements using Pierce™ BCA Protein Assay Kit (#23227, Thermo Scientific), and the rest was transferred to scintillation vials along with 3 ml scintillation fluid (OptiPhase HiSafe 3, #1200.437, PerkinElmer). Radioactivity was measured with a liquid scintillation counter (Tri-Carb 4910TR, PerkinElmer) and counts per minute were normalized to protein concentration.

## Flow cytometry

**FACS of human primary adipocyte progenitors for cell culture.** Human scWAT was digested and the stromal vascular fraction isolated as described above. Stromal vascular fraction was washed (0.5% BSA and 2 mmol/L EDTA in PBS), filtered through a 70 μm cell strainer, and incubated with fluorophore-conjugated antibodies (listed in Supplementary Table 1) in BD Horizon™ Brilliant Stain Buffer (#563794, BD Biosciences) for 30 min at 4 °C. Cells were again washed, filtered, and resuspended in FACS buffer (0.1% BSA (#A7030, Sigma-Aldrich), 2 mmol/L EDTA in PBS). To assess viability, cells were stained with 7-aminoactinomycin D (#559925, BD Biosciences) 15 min before FACS run. Fluorescence minus one (FMO) controls for all antibodies were

used to identify positive signals. The sorting was performed on a FACSAria™ Fusion instrument (BD Biosciences) equipped with 355, 405, 488, 561, and 633 nm lasers. Adipocyte progenitors were defined as CD45⁻/CD34⁺/CD31⁻ cells[93].

**FACS of immune cell populations in murine WAT**. Cryopreserved stromal vascular fraction cells from WT and IL1R1-KO mice were thawed, washed once with FACS buffer (PBS, 2% fetal calf serum and 4 mmol/L EDTA) and stained for 30 min with antibodies (Supplementary Table 1) together with a viability dye (Zombie Aqua™ Fixable Viability Kit, #423102, BioLegend) to discriminate between live and dead cells. Cells were then fixed using the eBioscience™ Foxp3/Transcription Factor Staining Buffer Set (#00-5523-00, Invitrogen) and acquired on a BD FACSymphony A5 (BD Biosciences) equipped with five lasers. Gating strategy is shown in Supplementary Fig. 10. Analyses were performed using FlowJo.

**FACS of progenitor subtypes in murine WAT**. Flow cytometry of stromal vascular fraction cells from WT and IL1R1-KO mice was done as described previously[94]. In brief, an equal number of stromal vascular fraction cells were transferred to designated tubes and washed with PBS. Non-specific binding was blocked with anti-mouse CD16/CD32 antibody (Miltenyi Biotec) in 1% BSA (in PBS) for 15 min at 4 °C. Cells were stained for 45 min at 4 °C (protected from light) with fluorescently labelled antibodies (Supplementary Table 1). Anti-mouse CD142 antibody was pre-conjugated with CF647 using the Biotium Mix-n-Stain CF647 Antibody Labeling Kit (#MX647S100, Sigma-Aldrich) per manufacturer's instructions. After washing two times, cells were finally resuspended in 1% BSA (in PBS). Dead cells were excluded with Dapi (Miltenyi Biotec). 30000 events were acquired on FACSCANTO 2 flow cytometer (BD Biosciences). All fluorophores were compensated with compensation controls, and gates were set with FMO controls. Cells were gated as described previously[95], based on their FSC-SSC appearance, followed by selection of single live CD45⁻/CD31⁻ (Lineage negative) cells and further separated into CD142⁺, CD142⁻/DPP4⁻/ICAM1⁺, or CD142⁻/ICAM1⁻/DPP4⁺ populations (gating strategy is shown in Supplementary Fig. 11). The flow cytometry data was analysed using FlowJo software and FACS Diva software v. 7.0 (BD Biosciences).

**Magnetic bead sorting of CD45⁺/⁻ stromal vascular fraction cells from mouse WAT**
Isolated stromal vascular fraction of scWAT and gWAT from WT and *ob/ob* mice were pelleted by centrifugation (200 *g*, 10 min, 4 °C) and a single cell suspension obtained using a 30 μm cell strainer. The cells were labelled with CD45 microbeads (#130-052-301, Miltenyi Biotec, Bergisch Gladbach, Germany) and magnetically separated using MS columns (#130-122-727, Miltenyi Biotec) or autoMACS™ Separator (Miltenyi Biotec), according to manufacturer's instructions.

**Quantification of lipid droplet accumulation**
Cells differentiated in 96-well plates were fixed with 4% paraformaldehyde for 10 min at room temperature, washed 3 times with PBS, stained with Hoechst 33342 (#H3570; 2 μg/ml) and Bodipy™ 493/503 (#D3922; 0.2 μg/ml) or HCS LipidTOX™ Red neutral lipid stain (#H34476; 1:5000) (all from Invitrogen, Thermo Fisher Scientific) for 20 min at room temperature, and washed three more times with PBS. Images of the cells were obtained by scanning the plate with a 10x objective using the high-content screening instrument CellInsight CX5 (Thermo Fisher Scientific). Nuclei and lipid droplets were quantified with the object and spot detection algorithms in the image analysis software HCS Studio: Cellomics Scan (v6.6.0) (Thermo Fisher Scientific). Nuclei at the border of the image frame were automatically excluded. Lipid droplet area per cell was calculated as the total area of lipid droplets in a well divided by the total number of nuclei in the same well. Post-scan and -quantification, the color, brightness and

contrast in the shown representative images were slightly modified (including gamma changes and pseudo-coloring) to improve visibility in print, using HCS Studio: Cellomics iView (v6.6.0) (Thermo Fisher Scientific). This processing was applied equally to the entire image and was identical in images from treatment and control. For single-cell resolution analyses, total lipid droplet area within the cell mask (mask type: circular; expanded by 40 pixels from nucleus mask) of each individual nucleus was used. Undifferentiated cells were defined as cells with a total lipid droplet area of maximum $5.324\,\mu m^2$ (corresponding to the first quartile of the lipid droplet area distribution of vehicle-treated cells).

## Time course of IL1R1 expression during in vitro differentiation
Gene expression of *IL1R1* in hASCs at different time points of differentiation was analyzed from a CAGE dataset within the FANTOM5 project[96]. hASCs from a 16-year-old male donor were expanded and induced to differentiate as described above. 5 μg of total RNA was used for CAGE library preparation, followed by sequencing on the HeliScope Single Molecule Sequencer platform, and mapping to the hg19 genome using Delve (https://fantom.gsc.riken.jp/software). Three replicates (defined as cells differentiated in separate wells and processed separately) per time point were used.

## RNA extraction, cDNA synthesis and qRT-PCR
Total RNA of murine WAT was extracted using RNeasy Lipid Tissue Mini Kit (#74804, Qiagen, Hilden, Germany). Freshly isolated murine adipocytes were frozen in Qiazol Reagent for RNA isolation using a modified protocol[97] for RNeasy Plus Universal Mini Kit (#73404, Qiagen, Hombrechtikon, Switzerland). RNA of non-adipocyte murine cells and hASCs was isolated using the NucleoSpin RNA kit (#740955, Macherey-Nagel, Bethlehem, PA) according to manufacturer's instructions. RNA concentration was measured using NanoDrop 2000 or NanoDrop One (Thermo Fisher Scientific). RNA was reverse transcribed using the iScript cDNA synthesis kit (#1708891, Bio-Rad Laboratories, Hercules, CA) and the thermo-cycler PTC100 (Bio-Rad Laboratories) or GoScript reverse transcriptase (#A2801, Promega, Madison, WI) and TProfessional TRIO Thermocycler (Analytik Jena). Real-time qRT-PCR was performed with TaqMan Universal PCR Master Mix (#43-181-57, Applied Biosystems, Thermo Fisher Scientific), iQ SYBR Green Supermix (#1708884, Bio-Rad Laboratories), or GoTaq qPCR (#A6002, Promega) assays on a QuantStudio 5 (Applied Biosystems) or ABI 7500 Fast Real-Time PCR System (Thermo Fisher Scientific) machine. Relative expression of each gene was calculated with the $2^{-\Delta\Delta CT}$ method using *PPIA* (human samples) or *gapdh* and *hprt* (murine samples) as reference genes. Probes and primers used are listed in Supplementary Table 1. Samples with undetectable levels of target mRNA were given a Ct value of 40 (Taqman) or 45 (SYBR Green) (corresponding to total number of cycles run) and are denoted with § in the figures.

## RNA-seq and downstream analyses
Isolated RNA from hASCs differentiated with or without IL-1β (10 ng/ml) and collected on day 2, 5, and 9 of differentiation was used for RNA-seq. The quality of total RNA was examined with Agilent Tapestation according to the manufacturer's instructions. Libraries were constructed from 200 ng total RNA using the Illumina TruSeq Stranded mRNA sample preparation protocol suitable for Illumina sequencing. The protocol included cDNA synthesis, ligation of adapters and amplification of indexed libraries. The yield and quality of the amplified libraries was analyzed using Qubit (Thermo Fisher) and Agilent Tapestation, respectively. The indexed cDNA libraries were normalized, combined, and the pool was sequenced on the Illumina Nextseq 550 for a 75-cycle v2 sequencing run, generating 75 base pair, paired end (42 + 42 cycles) with single index 6 bp. BCL conversion and demultiplexing of raw data was performed using bcl2fastq (v2.20.0).

Reads were assessed for quality by FastQC (v0.11.8) and aligned to the Ensembl GRCh38 reference genome using STAR (v2.6.1d). Gene counts were obtained using featureCounts (v1.5.1). Bioconductor package DESEq2 (v1.22.2) was used for normalization and sample group comparisons, generating log2 fold changes, Wald test p-values and p-values adjusted for multiple testing (Benjamini-Hochberg method). To identify IL-1β-regulated gene sets, a Gene Set Enrichment Analysis (GSEA)[98] was performed with the normalized counts from the RNA-seq data, using the software GSEA (v4.3.2) (Permutation type = gene set; Number of permutations = 1000; Enrichment statistic = weighted; Metric for ranking genes = Signal2Noise) and the Hallmark gene sets[99] from MSigDB (https://www.gsea-msigdb.org/gsea/msigdb). For volcano plots and GSEA, genes with a normalized count maxima of 5 were excluded. Prediction of IL-1β-regulated transcription factor motif activity changes was performed by ISMARA[100] on RNA-seq data collected on day 2 of differentiation, by uploading the fastq.gz files to ismara.unibas.ch followed by sample averaging.

## Western blot

Cells were lysed in Pierce™ RIPA buffer (#89901, Thermo Scientific) supplemented with protease and phosphatase inhibitors. Protein concentration was measured using Pierce™ BCA Protein Assay Kit (#23227, Thermo Scientific). 10-20 μg of protein was heated in a denaturation buffer at 95 °C for 5 min, and separated by SDS-PAGE electrophoresis. Proteins were transferred to PVDF membranes (Bio-Rad Laboratories), which were blocked for 1 h in 3% milk, and incubated at 4 °C overnight with primary antibodies. Membranes were washed in TBS-T and incubated with horseradish peroxidase-conjugated secondary antibodies at room temperature for 1 h. After additional washes with TBS-T, membranes were incubated with ECL Select™ Western Blotting Detection Reagent (Cytiva, Marlborough, MA) for 5 min and imaged in a ChemiDoc (Bio-Rad Laboratories). All antibodies used are listed in Supplementary Table 1. Uncropped and unprocessed blots are provided in the Source Data file.

## ChIP

ChIP assays were carried out using the MAGNA ChIP HiSens Chromatin Immunoprecipitation kit (#17-10460, Merck, Darmstadt, Germany) according to the manufacturer's protocol with minor alterations. Briefly, cells were cross-linked in 1% formaldehyde for 10 min followed by quenching with 137.5 mmol/L glycine for 5 min. After three washes with PBS, cells were scraped, pelleted, resuspended in lysis buffer (20 mmol/L Tris pH 8.0, 85 mmol/L KCl, 0.5% NP-40), incubated for 30 min at 4 °C, and homogenized by 20 needle strokes. After centrifugation, the pellet was resuspended in SCW buffer and sonicated with beads for 15 cycles (30 s ON/30 s OFF) in a Bioruptor® Pico (Diagenode, Liege, Belgium). Protein A/G magnetic beads were incubated with primary antibodies (listed in Supplementary Table 1) for ≥ 2 h at 4 °C and the sheared chromatin was then incubated with the bead-antibody mix at 4 °C overnight. After several washes, the pulled down chromatin was incubated in ChIP Elution Buffer with proteinase K (12 mAU/ml) at 65 °C for 2 h and 95 °C for 15 min, and then eluted from the beads. The pulled down DNA was used for qPCR (two technical replicates per sample, using the mean Ct-value) with the SYBR green assay, as described above (primers used are listed in Supplementary Table 1), and the % of total input of sheared chromatin from the matched sample was calculated.

## Analysis of postprandial transcriptomics data

A publicly available microarray dataset (GSE142401) of human scWAT from 19 middle-aged men[25] was used to analyze postprandial regulation of *IL1B*, *CEBPD*, and *CEBPB* by comparing quantile normalized expression values at baseline and 4 h after a meal in the dairy-based meal intervention group.

## Statistical analysis

Clinical cohort data was analyzed using IBM SPSS Statistics v28 (IBM SPSS, Armonk, NY). Normal distribution was assessed with the Kolmogorov-Smirnov test. Normally and non-normally distributed variables were assessed for correlation with Pearson's (data plotted together with linear regression line) or Spearman's correlation coefficients, respectively, and adjustments for BMI or body fat percent were performed with multiple regression analyses. Groups were compared by parametric unpaired or paired t test (two-tailed), non-parametric Mann-Whitney test (two-tailed) or Wilcoxon matched-pairs signed rank test (two-tailed), or one- or two-way ANOVA followed by Dunnett's, Šidák's or Tukey's multiple comparisons tests or Fisher's LSD tests, using GraphPad Prism v9 (GraphPad Software, San Diego, CA). This software was also used for making all graphical illustrations, except for bubble plots, which were generated using R. Box-and-whisker plots show the median (line), the first and third quartiles (box limits), and lowest and highest values (whisker ends). The lines of the violin plot represent median and quartiles. P-values < 0.05 were considered statistically significant.

## Reporting summary

Further information on research design is available in the Nature Portfolio Reporting Summary linked to this article.

## Data availability

RNA-seq data have been deposited at NCBI gene expression omnibus (GEO) under the accession number GSE237151. All data supporting the main findings are provided in the Source Data File and other data are available from the corresponding authors upon request. Additionally, previously published postprandial microarray data[25] can be accessed at GEO under the accession number GSE142401. Gene expression data of scWAT stromal vascular cell populations and paired mature adipocytes[90] is available at GEO under the accession number GSE100795. A CAGE dataset of hASCs at different time points of differentiation was previously collected[96] within the FANTOM5 project [https://fantom.gsc.riken.jp/5/data/]. Clinical cohort data have been previously published[46]. Source data are provided with this paper.

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

## Acknowledgements

This work was financially supported by research grants from the Swiss National Science Foundation (214900 and 212319) (M.Y.D), Swedish Research Council (2016–00694 (JL); 2020-01463 (I.W.A.)), Novo Nordisk Foundation (NNF210C0069989) (JL), the Strategic Research Programme in Diabetes at Karolinska Institute (JL), KID grant at Karolinska Institute (JL), EFSD/Novo Nordisk Programme for Diabetes Research in Europe 2023 (JL), the Swedish Diabetes Foundation (DIA2019-419) (I.W.A.), grant from the Swedish state under the agreement between the Swedish government and the county councils, the ALF-agreement (ALFGBG-990933) (I.W.A.), Diabetes Wellness Sweden (I.W.A), and Center for Innovative Medicine at Karolinska Institute (JL). We thank the team of the animal facility at the University Hospital Basel for taking excellent care of our laboratory animals. For valuable feedback on the project and/or manuscript, we thank S. Frendo-Cumbo (Karolinska Institute), A. de Baat (University of Basel and University Hospital Basel), as well as M. Aouadi (Karolinska Institute), who also, together with L. Levi (Karolinska Institute), provided WAT samples for the isolation of murine progenitors used for in vitro differentiation. We also thank J. Acosta and B. Tavira (Karolinska Institute) for technical help when initiating the project, and L. Steiger (University of Basel and University Hospital Basel) for supporting in vivo experiments. The ethical permit for experiments performed using human WAT stromal vascular fraction is held by prof. M. Rydén, who is head of the Endocrinology Unit at the Karolinska Institute where these experiments were performed, but not a co-author of the study. We thank the FACS facility of the Department of Biomedicine of University of Basel and MedH Flow Cytometry Core Facility at the Karolinska Institute. RNA-seq, including library construction, raw data processing and differential expression analyses, was performed at the Bioinformatics and Expression Analysis (BEA) Core Facility at the Karolinska Institute. Experimental schemes in Fig. 2 were created with Biorender.com.

## Author contributions

K.H. and J.P.S. performed and analyzed the experiments. K.H., J.P.S., D.T.M., M.B.-S., M.Y.D., and J.L. designed and advanced the project and wrote the manuscript. N.S., M.V., E.B., N.K.B. and I.W.A. helped with planning, performing, and/or analyzing flow cytometry experiments. L.R., H.M., C.Z., and E.D. performed in vivo experiments. All authors read, edited, and approved the final version of the manuscript.

## Funding

## Competing interests

The authors declare no competing interests.
