## [Peer Review File · Nature Communications]

IL-1 β promotes adipogenesis by directly targeting adipocyte precursorsREVIEWER COMMENTS

Reviewer #1 (Remarks to the Author):

In the present study, Hofwimmer and colleagues evaluated the potential physiological role of interleukin 1 beta (IL-1 β) in hyperplastic adipose tissue expansion and fat cell metabolism. By using different preclinical models and human samples, the authors found that IL-1 β promoted differentiation of adipocyte precursors in an acute manner, while chronically elevated levels of IL-1 β , as seen in obesity, desensitize progenitor cells to this beneficial effect of IL-1 β , thereby exacerbating inflammation. This is a timely, well-designed and interesting study showing a novel role of IL-1 β that confirms the concept of “healthy inflammation” in the adipose tissue that enables adipogenesis. Nonetheless, some specific points require to be amended.

Specific comments:

1. Introduction, lines 48-61: the impact of IL-1 β produced by white adipose tissue on glucose homeostasis is complex and it is also regulated by acute inflammation. In this sense, IL-1 β is an important mediator of adipocyte inflammatory cell death induced by SARS-CoV-2 adipose tissue infection, which triggers adipose tissue dysfunction and hyperglycemia in patients and preclinical models with COVID-19 (Reiterer M et al, Cell Metab 2021, PMID: 34599884; Saccon TD et al. Nature Commun 2022, PMID: 34599884).
2. The spatial mapping of human adipose tissue has identified 3 adipocyte progenitor cells, and one of the progenitor types clustered near macrophages and close to vascular and fibrotic tissues (Bäckdahl et al. Cell Metab 2021, PMID: 34380013). Additionally, the authors identified 3 different mature adipocytes that can be identified by marker genes encoding (LEP), perilipin-1 (PLIN1) and serum amyloid A (SAA), which exhibit specialized functions in terms of insulin sensitivity and leptin production and signalling (Bäckdahl et al. Cell Metab 2021, PMID: 34380013; Scheele C and Wolfrum C. Nat Rev Endocrinol 2021, PMID: 34667281). Thus, it would be very interesting to characterize the adipocyte progenitor cells that are differentiated to adipocytes in response to IL-1 β and/or the type of mature adipocyte arose from this stimulation.
3. During adipocyte differentiation, a rapid and transient increase in transcription and expression of C/EBP β and C/EBP δ is observed that activates the PPAR γ -induced expression of C/EBP α , which induces the transcription of adipocyte genes (LEP, FABP4 or SCD1, among others). In the late state of adipocyte differentiation, a rise of the transcription factor C/EBPZ is observed, which represents a negative regulator of adipogenesis (Darlington GJ, et al. J Biol Chem 1998, PMID: 9804754; Chen Y et al, Int J Obes 2022, PMID: 34789850). Thus, in order to show a complete framework of the impact of IL-1 β in adipocyte differentiation (Figure 6), it would be worth measuring changes in C/EBP α and C/EBPZ in response to IL-1 β .
4. Methods, lines 433-434: IL1R1-KO mice are on a C57BL/6J genetic background, while wild type treated with IL1Ra are on a C57BL/6N genetic background. There are small genetic differences between the C57BL/6J and C57BL/6N substrains leading to notable differences in the metabolic phenotypes (Fontaine DA et al. Diabetes 2016, PMID: 26696638; Kawashita E et al. Sci Reports 2019, PMID: 30659241). These differences might affect the results obtained in body weight gain and WAT morphology obtained after genetic

and pharmacological blockade of IL1R1 in the present study. Please discuss it.

5. Discussion: in order to gain further insight into the physiological relevance of the role of IL-1 β in adipogenesis, it would be interesting to mention the concept of healthy immunometabolism that was firstly proposed by Wernstedt Asterholm and colleagues (PMID: 24930973). This “healthy inflammation” scavenges the debris of dead adipocytes, enables adipogenesis as well as an appropriate extracellular matrix remodeling during the adipose tissue expansion.

Minor points

6. Methods, line 465: formalin is the commercial name of the chemical compound formaldehyde. Please use the chemical name.

7. Please use SI units (e.g. mol/L for molarity). The units of times gravity (g) must be written in italics.

8. Please use people-first language throughout the manuscript (e.g. subjects with obesity, instead of “obese humans”).

Reviewer #2 (Remarks to the Author):

Hofwimmer and colleagues attend to evaluate the potential role of IL1B in the control of adipogenesis by impacting adipocyte precursors. The raised hypothesis is novel but the concept is not always supported by the experimental results. The *in vivo* model developed appears not very well adapted to answer the main scientific question (progenitor cells must be targeted and not mature adipocytes).

Below my major concerns that must be experimentally addressed to be suitable for publication in Nature Communications.

Major comments:

- To evaluate the impact of IL1 pathway in adipocyte, IL1R1 AKO was developed but the phenotypic characterization of IL1R1 AKO upon CD or HFD was limited. The main conclusion is that IL1B signaling favor glucose uptake in FAT pads. This observation should be confirmed using additional methods as clamp.

- In contrary with the current study, many publications have reported that IL1B induce insulin resistance (Jager et al. Endocrinology 2007, Nov et al. Endocrinology 2010). This aspect should be clarified and discussed. Impact of acute IL1b treatment in lean versus obese mice should be explored? The biological of IL1B inhibition or treatment must be dependent of the surrounding adipose cells.

- The authors should also explain why the IL1B pathway (IL1R1) in adipocyte does not promote inflammation (Stat1 and Nf κ B signaling and promoter bindings should be measured).

- The manuscript seems to dissociate glucose uptake and adipogenesis from adipose tissue

inflammation. Inflammation has been proposed to be essential for physiological adipose tissue expansion (Zhu et al. *Molecular Metabolism* 2020 and Asterholm et al. *Cell metabolism* 2014). These publications (and related concept) should be better introduced.

- To evaluate the dialog between IL1b and AT expansion, authors used IL1R1 deletion in mature adipocytes (Cre under the control of adiponectin promoter). This model does not seem appropriate to address the scientific questions raised in the manuscript. Adipose progenitor cells should be targeted through the used PDGFRA-cre (constitutive or inducible).

- IL1R1 deletion provokes alteration in the proliferation of adipose progenitors. There are different subtypes of adipose progenitors i.e. intercellular adhesion molecule-1 (Icam1)+ committed progenitors, dipeptidyl peptidase-4+ (Dpp4 or CD26) Ly6c1+ early progenitors and Cd142+ adipogenesis-regulatory cells. These progenitors should be quantified to evaluate the effect of IL1B in specific subtypes of adipose progenitors.

- One very important finding is that gWAT is very sensitive to IL1B signaling while scWAT is less sensitive (lack of scWAT phenotype in IL1RA KO). Any explanation for this fat pad specific action?

- The human data are a bit struggling and not consistent with the mouse data. In fact, relevant associations between IL1R1 and Adipose markers are found in scWAT while the in vivo work has revealed a marked phenotype in the gWAT and not in scWAT.

- IL1b acute treatment increases C/EBP pathways. These data are very convincing. However, how IL1B increases C/EBP pathways is unclear. Deciphering this mechanism may help to strengthen the paper and could explain the discrepancy between g WAT and scWAT in mice upon DIO.

- The fact that IL1b inhibition may influence weigh gain is novel. The CANTOS study (Prof. Donath was one of the main investigators) may help to support this concept by looking at the body weight or waist circumference of participants before and after treatment.

- IL1B signaling is very important to polarize immune cells including adipose tissue macrophage and T cells (Th17 or MAIT-IL17 cells). Having the full characterization of adipose tissue immune cells may help to understand the unexpected phenotype observed during IL1B inhibition or acute stimulation?

Minor comments:

- Acute injection of IL1B: the concentration used is 1ug/kg. justification of this concentration should be provided. It does

Authors: We thank the reviewers for the thorough assessment of our manuscript and the positive feedback provided. We value the insightful and constructive comments, as well as the thoughtful criticisms, which have significantly contributed to the refinement of this study. In response to their queries, we have generated a substantial amount of new data. Below we provide point-by-point responses to reviewers' comments in italic. Text included into the manuscript is highlighted in blue.

REVIEWER COMMENTS

Reviewer #1 (Remarks to the Author):

In the present study, Hofwimmer and colleagues evaluated the potential physiological role of interleukin 1 beta (IL-1 β) in hyperplastic adipose tissue expansion and fat cell metabolism. By using different preclinical models and human samples, the authors found that IL-1 β promoted differentiation of adipocyte precursors in an acute manner, while chronically elevated levels of IL-1 β , as seen in obesity, desensitize progenitor cells to this beneficial effect of IL-1b, thereby exacerbating inflammation. This is a timely, well-designed and interesting study showing a novel role of IL-1 β that confirms the concept of “healthy inflammation” in the adipose tissue that enables adipogenesis. Nonetheless, some specific points require to be amended.

Authors: We thank the reviewer for this supportive feedback, especially the view that our work endorses the healthy inflammation concept, shedding new light on a novel role of IL-1 β as a promoter of adipogenesis. We addressed the major and minor comments as follows.

Specific comments:

- 1. Introduction, lines 48-61: the impact of IL-1 β produced by white adipose tissue on glucose homeostasis is complex and it is also regulated by acute inflammation. In this sense, IL-1 β is an important mediator of adipocyte inflammatory cell death induced by SARS-CoV-2 adipose tissue infection, which triggers adipose tissue dysfunction and hyperglycemia in patients and preclinical models with COVID-19 (Reiterer M et al, Cell Metab 2021, PMID: 34599884; Saccon TD et al. Nature Commun 2022, PMID: 34599884).**

Authors: We thank reviewer for the comment. The introduction of manuscript was rephrased, mentioning the involvement of IL-1b in adipocyte dysfunction upon SARS-Cov-2 infection.

INCLUDED IN THE INTRODUCTION: “A major pro-inflammatory factor involved in metabolic regulation is interleukin 1 β (IL-1 β). Recently, this cytokine was implicated in adipocyte dysfunction and cell death upon SARS-Cov-2 infection¹. This cytokine-IL-1 β is part of the obesity-associated chronic inflammation that can be detrimental to metabolic health².”

- 2. The spatial mapping of human adipose tissue has identified 3 adipocyte progenitor cells, and one of the progenitor types clustered near macrophages and close to vascular and fibrotic tissues (Bäckdahl et al. Cell Metab 2021, PMID: 34380013). Additionally, the authors identified 3 different mature adipocytes that can be identified by marker genes encoding (LEP), perilipin-1 (PLIN1) and serum**

amyloid A (SAA), which exhibit specialized functions in terms of insulin sensitivity and leptin production and signalling (Bäckdahl et al. Cell Metab 2021, PMID: 34380013; Scheele C and Wolfrum C. Nat Rev Endocrinol 2021, PMID: 34667281). Thus, it would be very interesting to characterize the adipocyte progenitor cells that are differentiated to adipocytes in response to IL-1 β and/or the type of mature adipocyte arose from this stimulation.

Authors: We thank the reviewer for this intriguing suggestion. In vitro systems of differentiated hASCs using the standard adipogenic cocktail is known to generate highly insulin-responsive adipocytes with high PLIN1 expression and very low leptin secretion. As such, our cell model appears to predominantly consist of the ADIPO^{PLIN1} subtype. This might reflect a more homogenous population of ASCs that are kept in culture for several passages or can be a result of the adipogenic cocktail used for in vitro differentiation, which pushes differentiation towards ADIPO^{PLIN1} subtype. Proper characterization of subpopulations requires single-cell techniques, which we believe would be beyond the scope of this paper, especially considering the concerns raised above. In addition, in our RNAseq data set we haven't observed an enrichment of genes specific for one of the three adipocyte subpopulations, suggesting that, at least in vitro, IL-1 β does not stimulate differentiation to a specific adipocyte subtype. Therefore, we instead focused on further characterization of progenitor subpopulations.

We performed flow cytometry sorting of fat tissue stromal vascular fraction using 9-week-old mice HFD-fed for 1 week. This experimental setup was chosen to match the timing of EdU treatment in our study. We show that deletion of IL-1R1 in mice do not alter the studied adipocyte progenitor populations (DPP4⁺ multipotent progenitors, ICAM1⁺ committed preadipocytes, and CD142⁺ regulatory progenitors, as described previously³) (Supplementary Fig. 3e-h).

Supplementary Fig. 3.

(e-h) Proportions of adipocyte progenitor subpopulations in scWAT (e, f) and gWAT (g, h) of 9-week-old IL1R1-KO and WT mice HFD-fed for 1 week (n = 6-8).

INCLUDED IN THE RESULTS: *No major changes in abundance of adipocyte progenitor populations were observed in IL1R1-KO mice (Supplementary Fig. 3e-h).*

Additionally, we FACS-sorted these three progenitor subpopulations from human stromal vascular fraction and differentiated them +/- IL-1 β in vitro (Figure 1 to reviewer). The results suggest that IL-1 β mainly targets CD142⁺, and possibly ICAM1⁺, but not DPP4⁺ progenitors. We believe that these results are beyond the focus of the current manuscript and have therefore chosen to not include them.

Figure 1 to Reviewer. Relative lipid droplet formation in sorted progenitor populations from human stromal vascular fraction, treated with IL-1 β or vehicle control for the first 6 days of differentiation and fixed and stained on day 14.

3. During adipocyte differentiation, a rapid and transient increase in transcription and expression of C/EBP β and C/EBP δ is observed that activates the PPAR γ -induced expression of C/EBP α , which induces the transcription of adipocyte genes (LEP, FABP4 or SCD1, among others). In the late state of adipocyte differentiation, a rise of the transcription factor C/EBPZ is observed, which represents a negative regulator of adipogenesis (Darlington GJ, et al. J Biol Chem 1998, PMID: 9804754; Chen Y et al, Int J Obes 2022, PMID: 34789850). Thus, in order to show a complete framework of the impact of IL-1 β in adipocyte differentiation (Figure 6), it would be worth measuring changes in C/EBP α and C/EBPZ in response to IL-1 β .

Authors: We thank the reviewer for this comment. Indeed, we missed to include C/EBP α and C/EBP ζ in the initial analysis. In the revised version we investigated the effect of IL-1 β on CEBA and DDIT3 (the gene encoding the anti-adipogenic protein known as CHOP or C/EBP ζ ⁴) expression during both early and late differentiation (as DDIT3 downregulation during early adipogenesis is important to allow progression of differentiation⁵). We observed that DDIT3 is downregulated on day 2 of differentiation and unaltered on day 9, which aligns with an early pro-adipogenic activity of IL-1 β . CEBA expression was unaltered on day 2 and downregulated on day 9. These additional results have been added to Fig. 5d and 5e.

Fig. 5.

d, e Adipogenesis- and adipocyte-related gene expression in hASCs treated with IL-1 β (10 ng/ml) on day 0 (**d**) or 7 (**e**) of differentiation and collected 48 h later (n = 3 independent experiments, 2 replicates per experiment).

INCLUDED IN THE RESULTS: Further, in a separate series of experiments where cells in distinct differentiation stages were treated with IL-1 β for the last 48 h prior to collection, genes

related to adipogenesis and mature adipocyte function were upregulated a gene expression pattern indicative of stimulated adipogenesis (including downregulation of the negative adipogenic regulator DDIT3⁵) was observed in early- (day 2) (Fig. 5d), but not late-stage (day 9) cells (Fig. 5e).

- 4. Methods, lines 433-434: IL1R1-KO mice are on a C57BL/6J genetic background, while wild type treated with IL1Ra are on a C57BL/6N genetic background. There are small genetic differences between the C57BL/6J and C57BL/6N substrains leading to notable differences in the metabolic phenotypes (Fontaine DA et al. Diabetes 2016, PMID: 26696638; Kawashita E et al. Sci Reports 2019, PMID: 30659241). These differences might affect the results obtained in body weight gain and WAT morphology obtained after genetic and pharmacological blockade of IL1R1 in the present study. Please discuss it.**

Authors: We agree with the reviewer that the C57BL/6J versus C57BL/6N genetic background has a significant effect on metabolism in part due to a mutation in C57BL/6J mice in the gene encoding nicotinamide nucleotide transhydrogenase (Nnt), a mitochondrial protein involved in the production of NADPH and reduction of mitochondrial reactive oxygen species⁶⁻⁹. This may indeed explain the elevated body weight and fat mass observed in HFD-fed C57BL/6J compared to C57BL/6N. The issue is now included in the discussion.

INCLUDED IN THE DISCUSSION: C57BL/6J mice, a strain associated to redox imbalances and metabolic alterations⁶⁻⁹, were used to investigate WAT morphology and adipogenesis in knockout animals, while WT C57BL/6N mice were employed for pharmacological IL-1 blockage. Despite recognized genetic and metabolic differences between the two sub-strains, we obtained robust results by using a large number of mice in three independent cohorts with age-, sex-, commercial vendor-, and genetic background-matched controls. ~~Curiously,~~ In contrast to our results in IL1R1-KO mice, IL-1Ra treatment in WT mice increased gonadal adipocyte formation. Although we cannot fully exclude the effect of genetic background, this discrepancy might reflect a true difference between transient pharmacological IL-1 blockage in adulthood and an inborn, chronic lack of IL-1 signaling, which may warrant further investigations.

- 5. Discussion: in order to gain further insight into the physiological relevance of the role of IL-1 β in adipogenesis, it would be interesting to mention the concept of healthy immunometabolism that was firstly proposed by Wernstedt Asterholm and colleagues (PMID: 24930973). This “healthy inflammation” scavenges the debris of dead adipocytes, enables adipogenesis as well as an appropriate extracellular matrix remodeling during the adipose tissue expansion.**

Authors: Thanks for the important remark. We agree with the reviewer that the mentioned publications should have been better discussed. They are now cited and discussed in the introduction and also in discussion sections.

INCLUDED IN THE DISCUSSION: Some years ago, the concept of healthy inflammation in adipose tissue emerged. Philipp Scherer’s group elegantly showed that inflammatory pathways

within adipocytes play a crucial role in facilitating adipose tissue expansion. The suppression of pro-inflammatory pathways in adipocytes led to adipose tissue dysfunction, ectopic lipid deposition, impaired glucose metabolism, and systemic inflammation. Moreover, the induction of acute inflammation in intact adipose tissue stimulated adipogenesis^{10, 11}. Our results complement the work of Zhu and Wernstedt-Asterholm et al, identifying IL-1 β as a key modulator of healthy inflammation as a modulator of adipose tissue expansion.

Minor points

- 6. Methods, line 465: formalin is the commercial name of the chemical compound formaldehyde. Please use the chemical name.** *The name has been corrected.*
- 7. Please use SI units (e.g. mol/L for molarity). The units of times gravity (g) must be written in italics.** *Addressed.*
- 8. Please use people-first language throughout the manuscript (e.g. subjects with obesity, instead of “obese humans”).** *This was corrected throughout the manuscript.*

Reviewer #2 (Remarks to the Author):

Hofwimmer and colleagues attend to evaluate the potential role of IL1B in the control of adipogenesis by impacting adipocyte precursors. The raised hypothesis is novel but the concept is not always supported by the experimental results. The in vivo model developed appears not very well adapted to answer the main scientific question (progenitor cells must be targeted and not mature adipocytes). Below my major concerns that must be experimentally addressed to be suitable for publication in Nature Communications.

Authors: We thank the reviewer for this the comprehensive critique, helpful considerations, and acknowledgment of the novelty of our work. We addressed the major and minor comments as follow:

Major comments:

- To evaluate the impact of IL1 pathway in adipocyte, ILR1 AKO was developed but the phenotypic characterization of IL1R1 AKO upon CD or HFD was limited. The main conclusion is that IL1B signaling favor glucose uptake in FAT pads. This observation should be confirmed using additional methods as clamp.

Authors: We value the insightful comment provided by the reviewer. In response to the noted limitation in phenotypic characterization of IL1R1^{AKO} mice, we performed additional experiments on male and female IL1R1^{AKO} mice (Supplementary Fig. 1a-h, m-q, u-x, and Supplementary Fig. 2a-c, h-k).

Regarding the IL-1 β -induced glucose uptake in fat pads (Fig. 1l-n), we have now confirmed this by additional glucose tracer experiments in vivo and in vitro. We opted for 'non-clamp' methodology as opposed to hyperglycaemic or hyperinsulinemic settings to avoid the potential confounding impact of insulin secretion/response that may mask the effect of IL-1 β on glucose uptake. As an alternative experiment to confirm IL-1 β -induced glucose uptake, we have adapted our assay and measured glucose uptake 18 min after simultaneous injection with the tracer and IL-1 β . In IL-1 β -treated mice, glucose uptake was elevated in gWAT while circulating levels of insulin levels were unaltered (Supplementary Fig. 2o-q). Additionally, we treated in vitro differentiated human adipocytes with IL-1 β , which also resulted in stimulated glucose uptake (Supplementary Fig. 2r). In line with our findings, other publications have shown IL-1 β -induced glucose uptake in WAT¹², 3T3-L1¹³, ovarian rat cells¹⁴, and synoviocytes¹⁵. We would like to emphasize that our two in vivo experimental setups utilize gold-standard techniques for measuring glucose uptake, relying on the detection of a radio-labeled glucose analog.

Thus, we believe that the issues raised in your constructive criticism have been adequately addressed.

INCLUDED IN THE RESULTS: *“Despite the notable ablation of Il1r1 in mature adipocytes of chow diet-fed IL1R1^{AKO} mice, no major changes were observed in mRNA levels of classical adipocyte or inflammatory markers; body weight development; food-intake, glucose, insulin, or IL-1 β levels during fasting-refeeding experiments; ~~not~~ insulin- or glucose tolerance (with or without IL-1 β challenge); adipocyte area; or tissue (including fat pads) weight compared to*

littermate controls (Fig. 1c-h and Supplementary Fig. 1a-x). Next, we metabolically challenged these female and male mice by high-fat diet (HFD) feeding starting at 8-10 weeks of age. IL1R1^{AKO} mice showed body weight, glucose tolerance, insulin levels; adipocyte area and mRNA markers; plasma glycerol; and tissue (including fat pads) weight similar to the littermate controls (Fig. 1e, i-k and Supplementary Fig. 2a-k)."

...

"Given that adipocytes in scWAT and gWAT express comparable levels of Il1r1 (Supplementary Fig. 2n), the reduced responsiveness to IL-1 β observed in scWAT may be due to a lower proportion of adipocytes in scWAT compared to gWAT¹⁶. The effect of IL-1 β on acute glucose uptake was further examined in an additional assay in WT mice. To avoid potential confounding effects of insulin, these mice were euthanized before an IL-1 β -stimulated increase in insulin secretion (Supplementary Fig. 2o). IL-1 β injection increased glucose uptake in gWAT also in this model (Supplementary Fig. 2p), although the effect in isolated adipocytes did not reach significance (Supplementary Fig. 2q). Furthermore, we observed an increased glucose uptake by acute IL-1 β treatment in human in vitro differentiated adipocytes (Supplementary Fig. 2r)."

Supplementary Fig. 1. Chow-fed IL1R1^{AKO} mice have no metabolic phenotype (related to Fig. 1).

(a-d) Food-intake (a), glucose (b), insulin (c), and IL-1 β (d) levels during fasting-refeeding experiment in 12-week-old chow-fed female mice (n = 9-12).

(e-h) Relative mRNA expression in scWAT (e), gWAT (f), mWAT (g), and brown adipose tissue (BAT) (h) of 19-20-week-old chow-fed male mice (n = 4-9).

(i-k) Insulin tolerance test (i), glucose tolerance test (j), and insulin levels (k) in 11-12- (i) and 12-13-week-old (j, k) male mice (n = 14-30).

(l, m) Concentration of circulating glucose and insulin during a glucose tolerance test in 12-13- (l) and 34-36-week-old (m) male chow-fed mice. An injection of saline or IL-1 β (1 μ g/kg bw) was administered 20 min before the glucose bolus.

(n-p) Insulin tolerance test (n), glucose tolerance test (o), and insulin levels (p) in 34-36- (n) and 35-36-week-old (o, p) male mice (n = 15-19).

(q) Concentration of circulating glucose during an insulin tolerance test in 46-48-week-old male chow-fed mice. An injection of saline or IL-1 β (1 μ g/kg bw) was administered 20 min before the insulin bolus (1.4 U/kg bw).

(r-t) Insulin tolerance test (r), glucose tolerance test (s), and insulin levels (t) in 48-50- (r) and 49-52-week-old (s, t) chow-fed male mice (n = 25-31).

(u-x) Cell size distribution (u), relative mRNA expression in adipocytes from gWAT (v), and organ and body weight (w, x) in 79-81-week-old chow-fed male mice (n = 9-11).

n = biological replicates. Data are shown as individual measurements and mean \pm SEM. *p < 0.05, **p < 0.01 by two-way ANOVA and Šidák's post-test (i-u) or unpaired nonparametric Mann-Whitney U test (all other panels).

Supplementary Fig. 2. Female HFD-fed IL1R1^{AKO} mice have no metabolic phenotype while HFD-fed IL1R1^{AKO} male mice show a mild insulin phenotype (related to Figure 1). (a-c) Body weight (a), insulin tolerance test (b), and glucose tolerance test (c) in 21-23- (a, c) and 19-21-week-old (b) HFD-fed female mice (n = 10). (d, e) Glucose tolerance test (d) and insulin levels (e) in 22-23-week-old HFD-fed male mice (n = 10-13). (f, g) Glucose tolerance test (f) and insulin levels (g) in 33-35-week-old HFD-fed male mice (n = 9-13).

(h-k) Cell size distribution (h), relative mRNA expression in adipocytes from gWAT (i), plasma glycerol (j) and organ weight (k) of 40-45-week-old HFD-fed male mice (n = 4-10).

(l, m) Insulin tolerance test in 21-22- (l) and 33-34-week-old (m) HFD-fed male mice (n = 10-13).

(n) Relative mRNA levels of *Il1r1* in adipocytes from gWAT and scWAT of 12-week-old male mice (n = 5).

(o-q) Insulin levels (o) and glucose uptake in gWAT (p) and gonadal adipocytes (q) of WT mice treated with saline or IL-1 β (1 μ g/kg bw) 18 min prior to euthanasia and tissue harvesting (n = 4-21).

(r) Glucose uptake in *in vitro* differentiated human adipose-derived stem cells treated with indicated concentrations of IL-1 β for 2 h (n = 4 independent experiments, 1-2 replicates per experiment).

HFD-feeding started at 10 weeks of age. n = biological replicates (a-q). Data are shown as individual measurements and mean \pm SEM. **p < 0.01, ****p < 0.0001 by unpaired nonparametric Mann-Whitney U test (a, j, k, n-q), one-way ANOVA and Tukey's multiple comparisons test (r), or two-way ANOVA and Šidák's post-test (all other panels).

- In contrary with the current study, many publications have reported that IL1B induce insulin resistance (Jager et al. Endocrinology 2007, Nov et al. Endocrinology 2010). This aspect should be clarified and discussed. Impact of acute IL1b treatment in lean versus obese mice should be explored? The biological of IL1B inhibition or treatment must be dependent of the surrounding adipose cells.

Authors: Thanks for bringing up this comment. The primary objective of our study is to highlight a novel physiological role of the cytokine IL-1 β , as a potentiator of adipogenesis. Our in vivo experimental setup, did not include IL-1 β treatment in lean nor obese mice because continuous administration of IL-1 β would most likely lead to pathological levels of this cytokine, triggering a maladaptive inflammatory response rather than stimulating adipogenesis, as suggested by Fig. 5j-l. It is important to note that the reported IL-1 β -induced insulin resistance in adipocytes mentioned by the reviewer is an outcome of chronic IL-1 β exposure (24-48 h of treatment). In contrast, our project focuses on the physiological actions of acutely elevated IL-1 β levels. However, we demonstrated that 48 h of IL-1 β pre-treatment suppressed the adipogenic potential of this cytokine, while exacerbating the expression of inflammatory markers (Fig. 5k-l). Notably, the adipogenic effect of IL-1 β was more pronounced when cells were treated with 2-h pulses of IL-1 β , mimicking postprandial IL-1 β exposure (Fig. j-k). Additionally, we have added experiments in which the effect of IL-1 β on lipid droplet accumulation was measured in immune cell-depleted SVF cells from ob/ob mice (same set-up as described for SVF from lean mice in Supplementary Fig. 5d-e). In contrast to lean mice (Supplementary Fig. 5d-e), IL-1 β was not able to promote lipid accumulation in the progenitors from ob/ob mice, in line with the notion that duration of IL-1 β exposure is an important factor, since progenitors from obese mice have likely been more chronically exposed to IL-1 β . Overall, our results support the notion that acute and chronic IL-1 β exposure lead to different biological effects in adipocytes and their progenitors. The discussion of manuscript was amended accordingly.

Supplementary Fig. 5.

(e, f) Lipid droplet accumulation (e) and cell number (f) in CD45⁻ stromal vascular cells from scWAT and gWAT of male *ob/ob* mice, differentiated with or without IL-1 β (10 ng/ml) (n = 3 independent experiments, 1-16 replicates per experiment).

*INCLUDED IN THE RESULTS: “In CD45-negative stromal vascular cells from lean mice, lipid droplet accumulation in response to IL-1 β was increased in differentiating ~~CD45-negative stromal vascular~~ cells from scWAT but reduced in gWAT-derived cells (Supplementary Fig. 5c), whereas cell number was slightly increased in cells from both depots (Supplementary Fig. 5d). Similarly, in gWAT-derived cells from *ob/ob* mice, IL-1 β reduced lipid droplet formation and increased cell number (Supplementary Fig. 5e, f). However, in contrast to cells from lean mice, IL-1 β had no effect on lipid droplet accumulation or cell number in scWAT-derived cells (Supplementary Fig. 5e, f). Taken together, these results suggest that IL-1 β , but not other inflammatory factors, potently promotes adipogenesis of human adipose-derived stem cells and that in murine precursors this effect is restricted to the subcutaneous depot ~~in murine precursors~~ of lean mice.”*

INCLUDED IN THE DISCUSSION: “The distinct adipogenic and inflammatory effects of IL-1 β on hASCs, depending on treatment timing and duration, may explain why some previous studies described an anti-adipogenic (and insulin-resistance) effect of IL-1 β ...”

- The authors should also explain why the IL1B pathway (IL1R1) in adipocyte does not promote inflammation (Stat1 and NfkB signaling and promoter bindings should be measured).

Authors: We thank the reviewer for this comment. We would like to clarify that IL-1 β signaling did promote inflammation in our experimental setup. Unlike the adipogenic effect, Fig. 5g, h illustrates that the inflammatory effect of IL-1 β is not limited to an early stage of differentiation. This observation is further confirmed by data depicted in Figure 5l, which demonstrates that IL-1 β treatment induces the expression of inflammatory markers across varying exposure durations. Moreover, prolonged IL-1 β exposure (pre-treatment, chronic exposure) exacerbated the expression of inflammatory markers. Thus, we believe that the points outlined above effectively address and clarify the reviewer comment.

- The manuscript seems to dissociate glucose uptake and adipogenesis from adipose tissue

inflammation. Inflammation has been proposed to be essential for physiological adipose tissue expansion (Zhu et al. Molecular Metabolism 2020 and Asterholm et al. Cell metabolism 2014). These publications (and related concept) should be better introduced.

Authors: Thanks for this important remark. We agree with the reviewer that the mentioned publications should have been better discussed. They were cited in the introduction in the original manuscript but are now also cited and more thoroughly discussed in the discussion section.

INCLUDED IN THE DISCUSSION: Some years ago, the concept of healthy inflammation in adipose tissue emerged. Philipp Scherer's group elegantly showed that inflammatory pathways within adipocytes play a crucial role in facilitating adipose tissue expansion. The suppression of pro-inflammatory pathways in adipocytes led to adipose tissue dysfunction, ectopic lipid deposition, impaired glucose metabolism, and systemic inflammation. Moreover, the induction of acute inflammation in intact adipose tissue stimulated adipogenesis^{10, 11}. Our results complement the work of Zhu and Wernstedt-Asterholm et al, identifying IL-1 β as a key modulator of healthy inflammation as a modulator of adipose tissue expansion.

- To evaluate the dialog between IL1b and AT expansion, authors used IL1R1 deletion in mature adipocytes (Cre under the control of adiponectin promoter). This model does not seem appropriate to address the scientific questions raised in the manuscript. Adipose progenitor cells should be targeted through the used PDGFRA-cre (constitutive or inducible).

Authors: We thank the reviewer for bringing attention to a very important aspect on targeting adipose progenitor cells. It is important to emphasize that while IL1R1^{AKO} mice were used for metabolic characterization and WAT morphology, these mice were not utilized in the investigation of new adipocyte formation. We employed global IL1R1-KO mice to study in vivo adipogenesis. Thus, in the mice used for the in vivo adipogenesis experiments, adipocyte progenitor cells were targeted and have no expression of IL1R1. Furthermore, there is currently no cre-driver mouse available to selectively target adipocyte progenitor cells. Pdgfra-Cre and Pdgfra-CreER mice have been used to target adipocyte progenitors as mentioned by the reviewer, but their lack of specificity is well-documented. Pdgfra expression has been reported in progenitor cells of heart^{17, 18}, testis¹⁹, and lungs²⁰, and whole-body Pdgfra knockout induces embryo lethality²¹. Several publications have shown that Pdgfra-Cre and Pdgfra-CreER induce Cre-mediated recombination in adipocyte progenitor cells but also in other cells types and tissues: oligodendrocytes in the central nervous system; stromal cells in bone marrow^{22, 23}; leukocytes (CD45⁺ cells) and endothelial cells in white adipose tissue^{23, 24}; trophoblast cells in the placenta²⁵; and Pdgfra-CreER-mediated recombination in liver, lung, spleen, kidney, and muscle^{22, 26}. Importantly, Pdgfra-CreER has been described as not useful for study of gene function in the adipocyte lineage, with low promoter efficiency, requesting high doses of tamoxifen to achieve partial recombination in adipocyte progenitor cells in scWAT and gWAT²⁶. It is important to emphasize that postnatal (p10) PDGRF α ⁺ cells but not PDGRF α ⁺ cells from adult mice (p60, 8 weeks of age) express adipose progenitor markers, indicating that PDGRF α

does not have a functional role in adult adipogenic potential²⁷. These data further restrict the utilization of this mouse model, as our experimental setup uses adult mice. Thus, it would be an advantage to specifically target progenitors, but there is to our knowledge no suitable model available. Given that adipocyte lineage cells were already targeted by the IL1R1-KO mice, we hope that the points mentioned above offer a comprehensive and satisfactory response to your criticism.

- IL1R1 deletion provokes alteration in the proliferation of adipose progenitors. There are different subtypes of adipose progenitors i.e. intercellular adhesion molecule-1 (Icam1)+ committed progenitors, dipeptidyl peptidase-4+ (Dpp4 or CD26) Ly6c1+ early progenitors and Cd142+ adipogenesis-regulatory cells. These progenitors should be quantified to evaluate the effect of IL1B in specific subtypes of adipose progenitors.

Authors: We thank the reviewer for this suggestion. In response, we performed the requested experiment, using 9-week-old mice HFD-fed for 1 week. We used this experimental setup because it matches with the timing of EdU treatment in our study. Deletion of IL-1R1 in mice did not alter the studied adipocyte progenitor populations (Supplementary Fig. 3g-j).

Supplementary Fig. 3.

(e-h) Proportions of adipocyte progenitor subpopulations in scWAT (e, f) and gWAT (g, h) of 9-week-old IL1R1-KO and WT mice HFD-fed for 1 week (n = 6-8).

Included in the results: No major changes in abundance of adipocyte progenitor populations were observed in IL1R1-KO mice (Supplementary Fig. 3e-h).

Additionally, we FACS-sorted these progenitor populations from human stromal vascular cells and differentiated them +/- IL-1 β in vitro (Figure 1 to Reviewer). The results suggest that IL-1 β mainly targets CD142⁺, and possibly ICAM1⁺, but not DPP4⁺ progenitors. We believe that these results are beyond the focus of the current manuscript and have therefore chosen to not include them.

Figure 1 to Reviewer. Lipid droplet formation in sorted progenitor populations from human stromal vascular cells, treated with IL-1 β or vehicle for the first 6 days of differentiation and fixed and stained on day 14.

- One very important finding is that gWAT is very sensitive to IL1B signaling while scWAT is less sensitive (lack of scWAT phenotype in IL1RA KO). Any explanation for this fat pad specific action?

*Authors: We thank the reviewer for raising this important point about IL-1 β response in different fat pads. To address this question, we measured *Il1r1* mRNA in adipocytes of gWAT and scWAT. Notably, adipocytes from both fat depots showed comparable expression levels of *IL1R1* (Supplementary Fig. 2n), indicating that *Il1r1* expression in adipocytes is not the determinant factor for the observed altered response to IL-1 β -induced glucose uptake. The heterogeneity of scWAT and gWAT is well documented. These fat depots comprise distinct cell populations (adipocytes, pre-adipocytes, immune cells, blood cells, endothelial cells, fibroblasts) and cellular lineages^{28, 29}. In gWAT, adipocytes constitute approximately 50% of the cellular composition, whereas only 30% of scWAT cells are adipocytes¹⁶. The reduced responsiveness to insulin and IL-1 β observed in scWAT during our glucose uptake assays is consistent with published data. Glucose uptake in subcutaneous adipocytes/WAT is reduced compared to “visceral” WAT (gonadal or omental), in humans and mice³⁰⁻³⁶. Therefore, the cellular composition of each depot likely plays a crucial role in the tissue response to insulin and IL-1 β . The elevated IL-1 β - and insulin-induced glucose uptake observed in gWAT could be attributed to the higher proportion of adipocytes in this fat depot, and this important point is now included in our results.*

*INCLUDED IN THE RESULTS: “There was no apparent effect of IL-1 β treatment in scWAT (Fig. 1n). Given that adipocytes in scWAT and gWAT express comparable levels of *Il1r1* (Supplementary Fig. 2n), the reduced responsiveness to IL-1 β observed in scWAT may be due to a lower proportion of adipocytes in scWAT compared to gWAT¹⁶.”*

- The human data are a bit struggling and not consistent with the mouse data. In fact, relevant associations between IL1R1 and Adipose markers are found in scWAT while the in vivo work has revealed a marked phenotype in the gWAT and not in scWAT.

Authors: We thank the reviewer for the comment. It is important to recognize the differences in the WAT morphology of humans and mice to avoid confusion during the translatability of research findings. In the human body, the primary fat storage depot is the scWAT, with the visceral WAT accounting for only 10-20% of the total body fat³⁷. In mice, approximately half of the total fat mass is concentrated within the abdominal cavity, with the greatest proportion found in the gWAT, serving as the primary depot in this region³⁷. Unlike humans, mice lack the fascia superficialis displaying a single layer in scWAT³⁷. Compared to human visceral WAT, human scWAT exhibits lower vascularity containing larger adipocytes with reduced mitochondrial content. Intriguingly, in mice, the inverse pattern is observed; gWAT exhibits increased vascularity, larger adipocytes, and reduced mitochondrial content compared to the subcutaneous counterpart³⁷. Therefore, despite the distinct anatomical locations, murine gWAT exhibits morphological similarities with human scWAT. This suggests that similar to scWAT in humans, gWAT in mice plays a significant role in storing excess energy in the form of triglycerides. Our in vivo results showed that effects of IL-1 on adipogenesis were stronger in gWAT compared to scWAT of mice, but comparisons of fat depots between different species is challenging and should consider not only the anatomical location but also morphological aspects. Nevertheless, our in vivo studies with mice and in vitro studies with human cells robustly showed that IL-1 signaling does potentiate adipogenesis. We believe that the points mentioned above offer a comprehensive response to your constructive comment.

- IL1b acute treatment increases C/EBP pathways. These data are very convincing. However, how IL1B increases C/EBP pathways is unclear. Deciphering this mechanism may help to strengthen the paper and could explain the discrepancy between g WAT and scWAT in mice upon DIO.

Authors: We thank the reviewer for this interesting suggestion that allowed us to gain some further mechanistic insights. Our hypothesis in the manuscript is that IL-1 β upregulates CEBPD (and possibly CEBPB indirectly) via CREB. Unfortunately, we were not able to find a CREB inhibitor that was able to block the IL-1 β -induced CREB activation. Instead, we pharmacologically inhibited the MAPKs Erk1/2, p38 and JNK in hASCs in vitro, since IL-1 β is a known activator of these pathways, and CREB is known to be activated by Erk and p38. Our data suggest that IL-1 β promotes adipogenesis via p38 (possibly partly in concert with Erk), likely through p38-mediated CREB activation and subsequent CEBPD upregulation (Fig. 6h-i)

Fig. 6.

h Western blot on whole-cell protein lysates collected 30 min after adipogenic induction with IL-1 β (10 ng/ml) and indicated MAPK inhibitor(s) (one representative blot from 3 independent experiments).

i CEBPD expression in hASCs 2 h after adipogenic induction in the presence of IL-1 β (10 ng/ml) and indicated MAPK inhibitor(s) (n = 3 independent experiments, 2 replicates per experiment).

j, k PPARG expression (**j**) and lipid droplet accumulation (**k**) after 24 h (**j**) and 9 days (**k**) of differentiation in hASCs treated with IL-1 β (10 ng/ml) and indicated MAPK inhibitor(s) for the first 2 and 24 h of differentiation, respectively (n = 3 independent experiments, 2 (**j**) and \geq 4 (**k**) replicates per experiment).

INCLUDED IN THE RESULTS: "IL-1 β is a known activator of the mitogen-activated protein kinases (MAPKs) extracellular signal-regulated kinase (ERK), p38 and c-Jun N-terminal kinase (JNK)³⁸, of which the two former pathways can activate CREB³⁹. Therefore, to elucidate through which signaling pathway IL-1 β activates CREB and upregulates CEBPD, we used pharmacological inhibitors to block each of these pathways during the first day of differentiation. ERK and p38 were also blocked simultaneously to avoid possible compensatory effects between these pathways on CREB activation. ERK or JNK inhibition alone did not affect the ability of IL-1 β to activate CREB (Fig. 6h). Conversely, IL-1 β -induced CREB phosphorylation was partially attenuated by p38 inhibition and completely abolished by simultaneous inhibition of p38 and ERK (Fig. 6h), suggesting that IL-1 β mainly activates CREB via p38 but that loss of this signaling pathway may be somewhat compensated for by ERK, but not vice versa. Similarly, the p38 inhibitor, both alone and in concert with ERK inhibition, also attenuated the IL-1 β -induced upregulation of CEBPD (Fig. 6i), supporting a role of CREB in this transactivation. Importantly, inhibiting p38, either alone or together with ERK, also attenuated IL-1 β -induced PPARG upregulation (Fig. 6j) and completely abolished IL-1 β -stimulated lipid droplet accumulation (Fig. 6k), supporting the involvement of p38-mediated CREB activation and subsequent CEBPD upregulation for the adipogenic effect of IL-1 β . Conversely, ERK inhibition alone blocked PPARG upregulation by IL-1 β (Fig. 6j) but failed to block its increase in CEBPD expression (Fig. 6i) and lipid droplet accumulation (Fig. 6k), whereas JNK inhibition attenuated IL-1 β -stimulated CEBPD upregulation (Fig. 6i) but not PPARG upregulation (Fig. 6j) or lipid droplet accumulation (Fig. 6k). Taken together, these results indicate that IL-1 β promotes adipogenesis through the p38 signaling pathway, likely partly via p38-mediated activation of CREB and subsequent upregulation of CEBPD."

- The fact that IL1b inhibition may influence weigh gain is novel. The CANTOS study (Prof. Donath was one of the main investigators) may help to support this concept by looking at the body weight or waist circumference of participants before and after treatment.

Authors: We thank the reviewer for this well-informed comment. Indeed, the finding of reduced body weight gain after IL-1Ra treatment is novel and not reported in the CANTOS study. However, this may be due to the low brain penetration of the anti-IL-1 β antibody used in this study.

Regarding pharmacological blockage of IL-1R1 in mice, the majority of the publications used inflammatory mouse models, utilizing short-term IL-1Ra treatment to mitigate such condition. In these studies, mice were euthanized immediately after the conclusion of the therapy. To the best of our knowledge, our study is the first to demonstrate body weight development in healthy WT HFD-fed mice for several weeks after cessation of IL-1Ra treatment, providing evidence for IL-1Ra-mediated reduction in body weight gain. It is worth noting that genetic deletion of caspase-1 or pharmacological blockage of NLRP3, IL-1-related molecules, reduced body weight gain in HFD-fed mice^{40, 41}. It is important to highlight how our experimental setup differs from clinical trials with human subjects. In our mouse model, the IL-1Ra treatment started two days before HFD-feeding. As a result, IL-1 inhibition initiates in lean healthy mice, preventing the onset of HFD-induced obesity. The effects of pharmacological IL-1 inhibition on adipogenesis in mice that are already obese remain to be studied.

- IL1B signaling is very important to polarize immune cells including adipose tissue macrophage and T cells (Th17 or MAIT-IL17 cells). Having the full characterization of adipose tissue immune cells may help to understand the unexpected phenotype observed during IL1B inhibition or acute stimulation?

Authors: To answer the question raised by the reviewer, we conducted an immune cell characterization of the stromal vascular fraction from scWAT and gWAT of 17-week-old mice HFD-fed for 9 weeks (Supplementary Fig. 3c-d), when body weight and WAT mass was significantly lower in IL1R1-KO mice (Fig. 2a-d). IL1R1-KO mice showed reduced levels of lipid-associated and non-perivascular macrophages in both fat pads, likely due to reduced IL-1 signaling and inflammation. Interestingly, IL1R1-KO mice revealed elevated levels of perivascular macrophages in gWAT. Dendritic cell depletion in mice was previously shown to cause resistance to obesity and normal body weight development⁴². Curiously, our IL1R1-KO mice showed elevation in abundance of a subset of dendritic cells in both fat depots, despite the reduced body weight. The levels of eosinophils, which have shown both positive and negative correlations with body weight⁴³, were increased in gWAT of IL1R1-KO mice. Th17 cell markers were not included in our immune cell panel due to the low frequency of this cell population. Alternatively, we measured Il17 mRNA in scWAT and gWAT of 9-week-old mice HFD-fed for 1 week. Il17 expression was not altered in IL1R1-KO mice (Figure 2 to Reviewer).

However, it should be noted that the CT values were high, near the limit of quantification of the method.

Supplementary Fig. 3.

(c, d) Proportions of immune cell subpopulations in scWAT (c) and gWAT (d) of 17-week-old HFD-fed IL1R1-KO and WT mice (n = 6-7).

Figure 2 to Reviewer.

Relative mRNA expression of *Il17* in scWAT (a) and gWAT (b) of 9-week-old IL1R1-KO and WT mice HFD-fed for 1 week (n = 4-5).

INCLUDED IN THE RESULTS: "In addition, IL1R1-KO mice showed alterations in the proportions of adipose tissue macrophages, dendritic cells, and eosinophils that might be coupled to body weight regulation (Supplementary Fig. 3c, d)."

Minor comments:

- Acute injection of IL1B: the concentration used is 1ug/kg. justification of this concentration should be provided.

Authors: Based on previous studies from our group, 1 ug/kg IL-1β was the concentration with stronger effect on glucose levels (Figure 3 to Reviewer).

Figure 3 to Reviewer. Justification of IL-1 β concentration.

(a) Circulating levels of IL-1 β 15, 30, and 60 min after intraperitoneal injection of 0.25, 0.5, and 1 $\mu\text{g}/\text{kg}$ bw IL-1 β .

(b) Effect of IL-1 β on glucose excursion. Concentration of circulating glucose and area under the curve during a glucose tolerance test. An injection of saline or IL-1 β (0, 0.25, 0.5, or 1 $\mu\text{g}/\text{kg}$ bw) was administered 20 min before the glucose bolus.

References

1. Saccon, T. D.; Mousovich-Neto, F.; Ludwig, R. G.; Carregari, V. C.; Dos Anjos Souza, A. B.; Dos Passos, A. S. C.; Martini, M. C.; Barbosa, P. P.; de Souza, G. F.; Muraro, S. P.; Forato, J.; Amorim, M. R.; Marques, R. E.; Veras, F. P.; Barreto, E.; Gonçalves, T. T.; Paiva, I. M.; Fazolini, N. P. B.; Onodera, C. M. K.; Martins Junior, R. B.; de Araújo, P. H. C.; Batah, S. S.; Viana, R. M. M.; de Melo, D. M.; Fabro, A. T.; Arruda, E.; Queiroz Cunha, F.; Cunha, T. M.; Pretti, M. A. M.; Smith, B. J.; Marques-Souza, H.; Knittel, T. L.; Ruiz, G. P.; Profeta, G. S.; Fontes-Cal, T. C. M.; Boroni, M.; Vinolo, M. A. R.; Farias, A. S.; Moraes-Vieira, P. M. M.; Bizzacchi, J. M. A.; Teesalu, T.; Chaim, F. D. M.; Cazzo, E.; Chaim, E. A.; Proença-Módena, J. L.; Martins-de-Souza, D.; Osako, M. K.; Leiria, L. O.; Mori, M. A., SARS-CoV-2 infects adipose tissue in a fat depot- and viral lineage-dependent manner. *Nat Commun* **2022**, *13* (1), 5722.
2. Rohm, T. V.; Meier, D. T.; Olefsky, J. M.; Donath, M. Y., Inflammation in obesity, diabetes, and related disorders. *Immunity* **2022**, *55* (1), 31-55.
3. Merrick, D.; Sakers, A.; Irgebay, Z.; Okada, C.; Calvert, C.; Morley, M. P.; Percec, I.; Seale, P., Identification of a mesenchymal progenitor cell hierarchy in adipose tissue. *Science* **2019**, *364* (6438).
4. Darlington, G. J.; Ross, S. E.; MacDougald, O. A., The role of C/EBP genes in adipocyte differentiation. *J Biol Chem* **1998**, *273* (46), 30057-60.
5. Batchvarova, N.; Wang, X. Z.; Ron, D., Inhibition of adipogenesis by the stress-induced protein CHOP (Gadd153). *Embo j* **1995**, *14* (19), 4654-61.
6. Close, A. F.; Chae, H.; Jonas, J. C., The lack of functional nicotinamide nucleotide transhydrogenase only moderately contributes to the impairment of glucose tolerance and glucose-stimulated insulin secretion in C57BL/6J vs C57BL/6N mice. *Diabetologia* **2021**, *64* (11), 2550-2561.
7. Fontaine, D. A.; Davis, D. B., Attention to Background Strain Is Essential for Metabolic Research: C57BL/6 and the International Knockout Mouse Consortium. *Diabetes* **2016**, *65* (1), 25-33.
8. Ronchi, J. A.; Figueira, T. R.; Ravagnani, F. G.; Oliveira, H. C.; Vercesi, A. E.; Castilho, R. F., A spontaneous mutation in the nicotinamide nucleotide transhydrogenase gene of C57BL/6J mice results in mitochondrial redox abnormalities. *Free Radic Biol Med* **2013**, *63*, 446-56.
9. Siersbæk, M. S.; Ditzel, N.; Hejbøl, E. K.; Præstholm, S. M.; Markussen, L. K.; Avolio, F.; Li, L.; Lehtonen, L.; Hansen, A. K.; Schrøder, H. D.; Krych, L.; Mandrup, S.; Langhorn, L.; Bollen, P.; Grøntved, L., C57BL/6J substrain differences in response to high-fat diet intervention. *Sci Rep* **2020**, *10* (1), 14052.
10. Wernstedt Asterholm, I.; Tao, C.; Morley, T. S.; Wang, Q. A.; Delgado-Lopez, F.; Wang, Z. V.; Scherer, P. E., Adipocyte inflammation is essential for healthy adipose tissue expansion and remodeling. *Cell Metab* **2014**, *20* (1), 103-18.
11. Zhu, Q.; An, Y. A.; Kim, M.; Zhang, Z.; Zhao, S.; Zhu, Y.; Asterholm, I. W.; Kusminski, C. M.; Scherer, P. E., Suppressing adipocyte inflammation promotes insulin resistance in mice. *Mol Metab* **2020**, *39*, 101010.
12. Dror, E.; Dalmas, E.; Meier, D. T.; Wueest, S.; Thevenet, J.; Thienel, C.; Timper, K.; Nordmann, T. M.; Traub, S.; Schulze, F.; Item, F.; Vallois, D.; Pattou, F.; Kerr-Conte, J.; Lavallard, V.; Berney, T.; Thorens, B.; Konrad, D.; Boni-Schnetzler, M.; Donath, M. Y., Postprandial macrophage-derived IL-1beta stimulates insulin, and both synergistically promote glucose disposal and inflammation. *Nat Immunol* **2017**, *18* (3), 283-292.
13. Jager, J.; Grémeaux, T.; Cormont, M.; Le Marchand-Brustel, Y.; Tanti, J. F., Interleukin-1beta-induced insulin resistance in adipocytes through down-regulation of insulin receptor substrate-1 expression. *Endocrinology* **2007**, *148* (1), 241-51.
14. Ben-Shlomo, I.; Kol, S.; Roeder, L. M.; Resnick, C. E.; Hurwitz, A.; Payne, D. W.; Adashi, E. Y., Interleukin (IL)-1beta increases glucose uptake and induces glycolysis in aerobically cultured rat ovarian cells: evidence that IL-1beta may mediate the gonadotropin-induced midcycle metabolic shift. *Endocrinology* **1997**, *138* (7), 2680-8.
15. Hervann, A.; Bourelly, B.; Le Maire, V.; Aussel, C.; Menkes, C. J.; Ekindjian, O. G., Action of anti-inflammatory drugs on interleukin-1 beta-mediated glucose uptake by synoviocytes. *Eur J Pharmacol* **1996**, *314* (1-2), 193-6.

16. Roh, H. C.; Tsai, L. T.; Lyubetskaya, A.; Tenen, D.; Kumari, M.; Rosen, E. D., Simultaneous Transcriptional and Epigenomic Profiling from Specific Cell Types within Heterogeneous Tissues In Vivo. *Cell Rep* **2017**, *18* (4), 1048-1061.
17. Chong, J. J.; Chandrakanthan, V.; Xaymardan, M.; Asli, N. S.; Li, J.; Ahmed, I.; Heffernan, C.; Menon, M. K.; Scarlett, C. J.; Rashidianfar, A.; Biben, C.; Zoellner, H.; Colvin, E. K.; Pimanda, J. E.; Biankin, A. V.; Zhou, B.; Pu, W. T.; Prall, O. W.; Harvey, R. P., Adult cardiac-resident MSC-like stem cells with a proepicardial origin. *Cell Stem Cell* **2011**, *9* (6), 527-40.
18. Tallquist, M. D.; Soriano, P., Cell autonomous requirement for PDGFRalpha in populations of cranial and cardiac neural crest cells. *Development* **2003**, *130* (3), 507-18.
19. Brennan, J.; Tilmann, C.; Capel, B., Pdgfr-alpha mediates testis cord organization and fetal Leydig cell development in the XY gonad. *Genes Dev* **2003**, *17* (6), 800-10.
20. Li, R.; Bernau, K.; Sandbo, N.; Gu, J.; Preissl, S.; Sun, X., Pdgfra marks a cellular lineage with distinct contributions to myofibroblasts in lung maturation and injury response. *Elife* **2018**, *7*.
21. Soriano, P., The PDGF alpha receptor is required for neural crest cell development and for normal patterning of the somites. *Development* **1997**, *124* (14), 2691-700.
22. O'Rourke, M.; Cullen, C. L.; Auderset, L.; Pitman, K. A.; Achatz, D.; Gasperini, R.; Young, K. M., Evaluating Tissue-Specific Recombination in a Pdgfralpha-CreERT2 Transgenic Mouse Line. *PLoS One* **2016**, *11* (9), e0162858.
23. Krueger, K. C.; Costa, M. J.; Du, H.; Feldman, B. J., Characterization of Cre recombinase activity for in vivo targeting of adipocyte precursor cells. *Stem Cell Reports* **2014**, *3* (6), 1147-58.
24. Berry, R.; Rodeheffer, M. S., Characterization of the adipocyte cellular lineage in vivo. *Nat Cell Biol* **2013**, *15* (3), 302-8.
25. Watez, J. S.; Qiao, L.; Lee, S.; Natale, D. R. C.; Shao, J., The platelet-derived growth factor receptor alpha promoter-directed expression of cre recombinase in mouse placenta. *Dev Dyn* **2019**, *248* (5), 363-374.
26. Jeffery, E.; Berry, R.; Church, C. D.; Yu, S.; Shook, B. A.; Horsley, V.; Rosen, E. D.; Rodeheffer, M. S., Characterization of Cre recombinase models for the study of adipose tissue. *Adipocyte* **2014**, *3* (3), 206-11.
27. Shin, S.; Pang, Y.; Park, J.; Liu, L.; Lukas, B. E.; Kim, S. H.; Kim, K. W.; Xu, P.; Berry, D. C.; Jiang, Y., Dynamic control of adipose tissue development and adult tissue homeostasis by platelet-derived growth factor receptor alpha. *Elife* **2020**, *9*.
28. Ferrero, R.; Rainer, P.; Deplancke, B., Toward a Consensus View of Mammalian Adipocyte Stem and Progenitor Cell Heterogeneity. *Trends Cell Biol* **2020**, *30* (12), 937-950.
29. Sanchez-Gurmaches, J.; Hung, C. M.; Guertin, D. A., Emerging Complexities in Adipocyte Origins and Identity. *Trends Cell Biol* **2016**, *26* (5), 313-326.
30. Virtanen, K. A.; Lonroth, P.; Parkkola, R.; Peltoniemi, P.; Asola, M.; Viljanen, T.; Tolvanen, T.; Knuuti, J.; Ronnema, T.; Huupponen, R.; Nuutila, P., Glucose uptake and perfusion in subcutaneous and visceral adipose tissue during insulin stimulation in nonobese and obese humans. *J Clin Endocrinol Metab* **2002**, *87* (8), 3902-10.
31. Bashan, N.; Dorfman, K.; Tarnovscki, T.; Harman-Boehm, I.; Liberty, I. F.; Bluher, M.; Ovadia, S.; Maymon-Zilberstein, T.; Potashnik, R.; Stumvoll, M.; Avinoach, E.; Rudich, A., Mitogen-activated protein kinases, inhibitory-kappaB kinase, and insulin signaling in human omental versus subcutaneous adipose tissue in obesity. *Endocrinology* **2007**, *148* (6), 2955-62.
32. Stolic, M.; Russell, A.; Hutley, L.; Fielding, G.; Hay, J.; MacDonald, G.; Whitehead, J.; Prins, J., Glucose uptake and insulin action in human adipose tissue--influence of BMI, anatomical depot and body fat distribution. *Int J Obes Relat Metab Disord* **2002**, *26* (1), 17-23.
33. Lundgren, M.; Buren, J.; Ruge, T.; Myrnas, T.; Eriksson, J. W., Glucocorticoids down-regulate glucose uptake capacity and insulin-signaling proteins in omental but not subcutaneous human adipocytes. *J Clin Endocrinol Metab* **2004**, *89* (6), 2989-97.
34. Westergren, H.; Danielsson, A.; Nystrom, F. H.; Stralfors, P., Glucose transport is equally sensitive to insulin stimulation, but basal and insulin-stimulated transport is higher, in human omental compared with subcutaneous adipocytes. *Metabolism* **2005**, *54* (6), 781-5.
35. Kovsan, J.; Osnis, A.; Maisel, A.; Mazor, L.; Tarnovscki, T.; Hollander, L.; Ovadia, S.; Meier, B.; Klein, J.; Bashan, N.; Rudich, A., Depot-specific adipocyte cell lines reveal differential

- drug-induced responses of white adipocytes--relevance for partial lipodystrophy. *Am J Physiol Endocrinol Metab* **2009**, *296* (2), E315-22.
36. Christen, T.; Sheikine, Y.; Rocha, V. Z.; Hurwitz, S.; Goldfine, A. B.; Di Carli, M.; Libby, P., Increased glucose uptake in visceral versus subcutaneous adipose tissue revealed by PET imaging. *JACC Cardiovasc Imaging* **2010**, *3* (8), 843-51.
37. Hagberg, C. E.; Spalding, K. L., White adipocyte dysfunction and obesity-associated pathologies in humans. *Nat Rev Mol Cell Biol* **2023**.
38. Kumar, A.; Middleton, A.; Chambers, T. C.; Mehta, K. D., Differential roles of extracellular signal-regulated kinase-1/2 and p38(MAPK) in interleukin-1beta- and tumor necrosis factor-alpha-induced low density lipoprotein receptor expression in HepG2 cells. *J Biol Chem* **1998**, *273* (25), 15742-8.
39. Wiggin, G. R.; Soloaga, A.; Foster, J. M.; Murray-Tait, V.; Cohen, P.; Arthur, J. S., MSK1 and MSK2 are required for the mitogen- and stress-induced phosphorylation of CREB and ATF1 in fibroblasts. *Mol Cell Biol* **2002**, *22* (8), 2871-81.
40. Stienstra, R.; Joosten, L. A.; Koenen, T.; van Tits, B.; van Diepen, J. A.; van den Berg, S. A.; Rensen, P. C.; Voshol, P. J.; Fantuzzi, G.; Hijmans, A.; Kersten, S.; Müller, M.; van den Berg, W. B.; van Rooijen, N.; Wabitsch, M.; Kullberg, B. J.; van der Meer, J. W.; Kanneganti, T.; Tack, C. J.; Netea, M. G., The inflammasome-mediated caspase-1 activation controls adipocyte differentiation and insulin sensitivity. *Cell Metab* **2010**, *12* (6), 593-605.
41. Thornton, P.; Reader, V.; Digby, Z.; Smolak, P.; Lindsay, N.; Harrison, D.; Clarke, N.; Watt, A. P., Reversal of High Fat Diet-Induced Obesity, Systemic Inflammation, and Astroglialosis by the NLRP3 Inflammasome Inhibitors NT-0249 and NT-0796. *J Pharmacol Exp Ther* **2024**, *388* (3), 813-826.
42. Soedono, S.; Cho, K. W., Adipose Tissue Dendritic Cells: Critical Regulators of Obesity-Induced Inflammation and Insulin Resistance. *Int J Mol Sci* **2021**, *22* (16).
43. Calco, G. N.; Fryer, A. D.; Nie, Z., Unraveling the connection between eosinophils and obesity. *J Leukoc Biol* **2020**, *108* (1), 123-128.

REVIEWERS' COMMENTS

Reviewer #1 (Remarks to the Author):

I thank the Authors for addressing all my comments/suggestions. The manuscript has been substantially improved after revision, and it is my belief that it is now ready for publication in Nature Communications.

Reviewer #2 (Remarks to the Author):

The authors have adequately addressed all of my previous concerns. I congratulate them for producing a significant amount of data that better support their conclusions. The incorporated amendments assisted in generating a stronger, clearer and improved manuscript. The authors also highlight the relevance of their results to human adipose tissue, demonstrating the importance of IL1 signalling in human obesity.

This study is novel and timely as it elegantly characterizes the temporal and spatial dynamics of IL1 signalling and adipose tissue expansion. It also illuminates a crucial role for IL1 pathways in diabetes and may help to better understand the "non-canonical" functions of IL1B.

Altogether, I recommend to accept the manuscript as it is. Congratulations to the authors on the completion of this study